# Where Rectified Flows Leak:
# Characterising Membership Signals Along the Interpolation Path

**Thomas Sesmat** [* 1]  **Gabriel Meseguer-Brocal** [2]  **Geoffroy Peeters** [1]

## Abstract

Understanding what generative models retain from training data remains challenging, with implications for copyright and privacy. Beyond verbatim reproduction, models can encode subtler traces of their training data that never surface in their outputs yet remain exploitable. We study this regime for Rectified Flows, which are increasingly used in deployed generative systems. We analyse the interpolation path $X_\lambda = (1 - \lambda)X_0 + \lambda X_1$ that defines the Rectified Flow training. We show that a gap exists between the reconstruction of train and test data that follows a bell-shaped curve over $\lambda$, wich accumulates during training, while the validation metrics remain stable. The signal has a maximum whose location we derive in closed form under Gaussian assumptions. We validate these predictions on both audio and images and show that the bell-shaped structure is universal, while the peak prediction holds when our assumptions are satisfied. As a proof of concept, we exploit this specific $\lambda$-resolved structure to perform a Membership Inference Attack, distinguishing members of the training set from non-members.

## 1. Introduction

The deployment of generative models has raised legal concerns across multiple domains. Lawsuits have been filed over unauthorised use and direct reproduction of copyrighted photographs (cou, 2023; Somepalli et al., 2023), text from news organisations and authors (cou, 2023), and music from major record labels (Recording Industry Association of America, 2024; Newton-Rex, 2024). Beyond verbatim reproduction lies a spectrum of subtler forms of

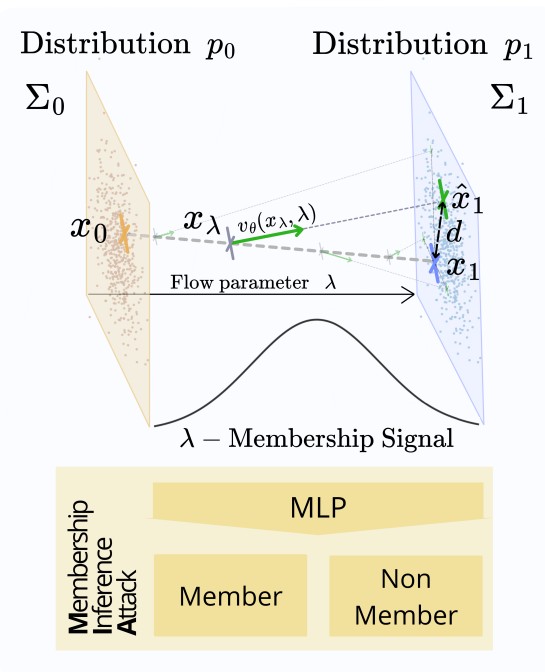

*Figure 1.* Overview of our approach. **Top:** Detection protocol, given a sample $x_1$, we interpolate with noise $x_0$ at varying $\lambda$, predict the velocity $v_\theta(x_\lambda, \lambda)$, and measure reconstruction error $d = \|x_1 - \hat{x}_1\|^2$. **Middle:** The train-test gap in reconstruction error follows a bell-shaped curve over $\lambda$; we derive a closed-form expression for the peak location. **Bottom:** As a proof of concept, the $\lambda$-resolved errors can be fed to an MLP classifier to perform Membership Inference Attack.

memorisation: a trained model may reconstruct training samples more accurately, respond more confidently near them, or otherwise treat them differently from held-out data, all without ever reproducing them. We refer to such measurable asymmetries as the **membership signal**. We study its structure in Rectified Flows (Liu et al., 2023; Lipman et al., 2023), which underlie widely deployed systems such as FLUX.1 (Black Forest Labs, 2024), VoiceBox (Le et al., 2023), and Stable Audio Open (Evans et al., 2024). Our analysis focuses on the structural properties of the framework rather than on attacks against specific deployed models.

Characterising the membership signal is challenging because aggregate training metrics offer little guidance: a

[1]LTCI, Télécom Paris, Institut Polytechnique de Paris, Palaiseau, France [2]Deezer Research, Paris, France. Correspondence to: Thomas Sesmat <thomas.sesmat@telecom-paris.fr>.

*Proceedings of the $43^{rd}$ International Conference on Machine Learning*, Seoul, South Korea. PMLR 306, 2026. Copyright 2026 by the author(s).

model can encode rich information about its training data while its loss curves show no sign of overfitting (Tirumala et al., 2022). Where, in the model's behaviour, does this information reside? Existing studies of memorisation in diffusion models suggest that intermediate timesteps carry most of the signal (Matsumoto et al., 2023). A theoretical understanding of where and why the signal concentrates remains lacking, particularly for Rectified Flows, whose deterministic interpolation path differs from iterative denoising.

We propose to characterise the membership signal along the interpolation path $X_\lambda = (1 - \lambda)X_0 + \lambda X_1$ that defines Rectified Flow training. This path offers a continuum of positions to analyse how the model treats training versus held-out data: at $\lambda = 0$, the model observes pure noise; at $\lambda = 1$, the data itself. The intermediate regime is where the model must leverage learnt structure to predict the velocity and where membership signals emerge. We illustrate our approach in Figure 1.

**Paper organisation and contributions.** After reviewing related works in Section 2, we demonstrate mathematically in Sections 3 and 4 that a gap exists between the reconstruction of train and test data that follows a bell-shaped curve over $\lambda$. We derive a closed-form expression for the peak location $\lambda_F^*$ as a function of the covariances $\Sigma_0$ and $\Sigma_1$, identifying where the membership signal is maximal. In Sections 5 and 6, we validate these theoretical predictions experimentally on various modalities (audio and image datasets), latent spaces, architectures, and noise configurations. We show that the bell-shaped structure is universal while the peak prediction holds when our Gaussian assumptions are satisfied. Finally, in Section 7, we demonstrate, as a proof of concept, that this $\lambda$-resolved structure is exploitable by a simple Membership Inference Attack (MIA, distinguishing training from held-out samples) on a piano music dataset. For reproducibility, our experimental code is available here.

## 2. Related Work

**Rectified Flows.** Rectified Flows (Liu et al., 2023) and Flow Matching (Lipman et al., 2023) learn velocity fields via regression on linearly interpolated samples $X_\lambda = (1 - \lambda)X_0 + \lambda X_1$. Unlike diffusion models that require many denoising steps, Rectified Flows learn straighter paths between noise and data, enabling high-quality generation in fewer steps. This efficiency has driven adoption in major systems including Stable Diffusion 3 (Esser et al., 2024), FLUX (Black Forest Labs, 2024), and Stable Audio Open (Evans et al., 2024). Liu et al. (2023) also introduces a reflow procedure that further straightens trajectories by iterating training through the learnt velocity field, replacing the independent coupling between $X_0$ and $X_1$ with a learnt pairing.

**Memorisation in generative models.** The most studied form of memorisation is verbatim reproduction, where models regenerate training samples exactly (Carlini et al., 2023; Somepalli et al., 2023). For diffusion models, Bonnaire et al. (2025) characterise this phenomenon through two timescales: $\tau_{\mathrm{gen}}$ at which quality generation begins, and $\tau_{\mathrm{mem}}$ beyond which memorisation emerges. Gu et al. (2025) systematically studies factors affecting such memorisation, including dataset size, model capacity, and the surprising role of random labels. For Flow Matching, Gao & Li (2024) derives analytical expressions for the optimal velocity field and analyses memorisation in sample data subspaces, while Bertrand et al. (2025) identifies distinct temporal phases in the generative process.

Ippolito et al. (2023) argues that memorisation exists on a spectrum of similarity to training data, ranging from exact reproduction to subtle statistical traces. Crucially, preventing verbatim reproduction does not eliminate the risk: models can still leak information through paraphrase, stylistic similarity, or structural patterns. Alternative definitions formalise this intuition, such as counterfactual memorisation, which measures how predictions change when a specific sample is removed from training (Zhang et al., 2023).

At the subtle end of this spectrum lies train-test distinguishability: a model may produce novel samples while still encoding exploitable signals about its training data. We refer to this measurable asymmetry as the **membership signal**, and it is the form of memorisation we study. It remains comparatively underexplored: Tirumala et al. (2022) shows that it can occur without visible overfitting on loss curves. Feldman (2020) argues that some memorisation is necessary for generalisation on long-tailed distributions, suggesting it is not inherently undesirable but rather a phenomenon to understand.

**Trajectory-dependent memorisation signals** The observation that memorisation signals depend on position along the denoising trajectory is not new. Matsumoto et al. (2023) report that intermediate timesteps are the most vulnerable to MIA, with success varying predictably across the denoising trajectory. Other MIA methods developed for diffusion models, such as SecMI (Duan et al., 2023) and PIA (Kong et al., 2023), also leverage trajectory information, though they rely on the iterative denoising structure and do not transfer directly to Rectified Flows. More broadly, Shokri et al. (2017) formalised the MIA as a diagnostic for studying what models retain. Our work extends this trajectory perspective to Rectified Flows and grounds it theoretically: we derive why the membership signal peaks at a specific location $\lambda_F$ determined by data statistics, rather than discovering it empirically.

# 3. Mathematical Setup

We establish the framework for analysing membership signals in latent Rectified Flows. For readability, all proofs are deferred to Appendix A.

## 3.1. Distributions and Interpolation

Let $X_0 \sim p_0 = \mathcal{N}(0, \Sigma_0)$ denote samples from a noise distribution and $X_1 \sim p_1$ denote latent representations of data, with covariance $\Sigma_1$. We assume $X_0 \perp\!\!\!\perp X_1$, which holds by construction in Rectified Flow training without reflow (Liu et al., 2023). Define:

$$X_\lambda = (1 - \lambda)X_0 + \lambda X_1 \qquad \text{(interpolation)} \quad (1)$$
$$V = X_1 - X_0 \qquad \text{(velocity)} \quad (2)$$

The optimal predictor is the conditional expectation:

$$v^*(x, \lambda) \triangleq \mathbb{E}_{p_0 \times p_1}[V \mid X_\lambda = x] \quad (3)$$

This is a deterministic function fully determined by $(p_0, p_1)$.

By the definition of conditional expectation, $\mathbb{E}_{p_0 \times p_1}[V - v^*(X_\lambda, \lambda) \mid X_\lambda] = 0$. This orthogonality property implies that for any measurable function $g : \mathbb{R}^d \to \mathbb{R}^d$:

$$\mathbb{E}_{p_0 \times p_1}[\langle g(X_\lambda), V - v^*(X_\lambda, \lambda)\rangle] = 0 \quad (4)$$

The irreducible variance is:

$$\sigma_{\text{irr}}^2(\lambda) \triangleq \mathbb{E}_{p_0 \times p_1}[\|V - v^*(X_\lambda, \lambda)\|^2] \quad (5)$$

This quantity depends only on the distributions $(p_0, p_1)$ and represents a fundamental limit: since $v^*$ is the optimal predictor, no model can achieve a lower expected squared error, regardless of its capacity.

## 3.2. Training and Test Sets

Let $\mathcal{D}^{\text{train}} = \{(x_0^{(i)}, x_1^{(i)})\}_{i=1}^n$ be a training set drawn i.i.d. from $p_0 \times p_1$. For each sample $i \in \{1, \dots, n\}$, define:

$$v^{(i)} \triangleq x_1^{(i)} - x_0^{(i)} \quad (6)$$
$$x_\lambda^{(i)} \triangleq (1 - \lambda)x_0^{(i)} + \lambda x_1^{(i)} \quad (7)$$
$$\epsilon_i(\lambda) \triangleq v^{(i)} - v^*(x_\lambda^{(i)}, \lambda) \quad (8)$$

Once $\mathcal{D}^{\text{train}}$ is drawn, these are fixed vectors in $\mathbb{R}^d$.

Let $\mathcal{D}^{\text{test}} = \{(\tilde{x}_0^{(j)}, \tilde{x}_1^{(j)})\}_{j=1}^m$ be a test set drawn i.i.d. from $p_0 \times p_1$, independently of $\mathcal{D}^{\text{train}}$. Define $\tilde{v}^{(j)}$, $\tilde{x}_\lambda^{(j)}$, and $\tilde{\epsilon}_j(\lambda)$ analogously.

A model $v_\theta : \mathbb{R}^d \times [0, 1] \to \mathbb{R}^d$ is trained on $\mathcal{D}^{\text{train}}$. The parameter $\theta$ depends on both the training and the randomness in the training procedure (initialisation, batch ordering, etc.). In the following analysis, we condition on the trained model: once $\theta$ is fixed, $v_\theta$ is a deterministic function.

## 3.3. Loss Decomposition and the Membership Signal

The training loss $L^{\text{train}}(\lambda) = \frac{1}{n} \sum_{i=1}^n \|v_\theta(x_\lambda^{(i)}, \lambda) - v^{(i)}\|^2$ decomposes as:

$$L^{\text{train}}(\lambda) = E_n^{\text{train}}(\lambda) + \hat{\sigma}_n^2(\lambda) - 2G_n^{\text{train}}(\lambda) \quad (9)$$

where $E_n^{\text{train}} = \frac{1}{n} \sum_i \|v_\theta(x_\lambda^{(i)}, \lambda) - v^*(x_\lambda^{(i)}, \lambda)\|^2$ is the empirical approximation error, $\hat{\sigma}_n^2 = \frac{1}{n} \sum_i \|\epsilon_i(\lambda)\|^2$ the empirical irreducible variance, and:

$$G_n^{\text{train}}(\lambda) \triangleq \frac{1}{n} \sum_{i=1}^n \langle v_\theta(x_\lambda^{(i)}, \lambda) - v^*(x_\lambda^{(i)}, \lambda), \epsilon_i(\lambda)\rangle \quad (10)$$

The test loss admits the same decomposition with analogous terms $E_m^{\text{test}}$, $\hat{\sigma}_m^2$, and $G_m^{\text{test}}$. The difference lies in the cross-correlation term:

**Proposition 3.1** (Train-test asymmetry). *Conditioned on the training set $\mathcal{D}^{\text{train}}$:*

$$\mathbb{E}_{\mathcal{D}^{\text{test}}}[G_m^{\text{test}}(\lambda) \mid \mathcal{D}^{\text{train}}] = 0 \quad (11)$$

*whereas $G_n^{\text{train}}(\lambda)$ on training data is a priori generically non-zero.*

To isolate $G_n^{\text{train}}(\lambda)$ from other terms in the train-test gap, we introduce two assumptions.

**Assumption 3.2** (Uniform approximation error). The model's deviation from $v^*$ is the same on training points as on the population:

$$E_n^{\text{train}}(\lambda) = E^{\text{pop}}(\lambda) \triangleq \mathbb{E}_{p_0 \times p_1}[\|v_\theta(X_\lambda, \lambda) - v^*(X_\lambda, \lambda)\|^2] \quad (12)$$

This holds when the model has not overfit in the classical sense, e.g., thanks to early stopping. Note that this does not preclude a train-test gap in the loss, which can arise through $G_n^{\text{train}}(\lambda)$.

**Assumption 3.3** (Representative sample). The empirical irreducible variance matches its population value:

$$\hat{\sigma}_n^2(\lambda) = \sigma_{\text{irr}}^2(\lambda) \quad (13)$$

This holds by the law of large numbers for large $n$.

Under these assumptions, conditioning on the training set $\mathcal{D}^{\text{train}}$ (and hence on the trained model $v_\theta$), the expected train-test gap over fresh test samples reduces to:

$$\mathbb{E}_{\mathcal{D}^{\text{test}}}[\Delta(\lambda) \mid \mathcal{D}^{\text{train}}] = 2G_n^{\text{train}}(\lambda) \quad (14)$$

The quantity $G_n^{\text{train}}(\lambda)$ is the **membership signal**: it measures the correlation between the model's deviation from $v^*$ and the training-specific residuals $\epsilon_i(\lambda)$.

## 3.4. Covariance Structure

Under $X_0 \perp\!\!\!\perp X_1$, direct computation yields:

$$\Phi(\lambda) \triangleq \text{Cov}(X_\lambda) = (1-\lambda)^2 \Sigma_0 + \lambda^2 \Sigma_1 \qquad (15)$$

$$C(\lambda) \triangleq \text{Cov}(V, X_\lambda) = \lambda \Sigma_1 - (1-\lambda)\Sigma_0 \qquad (16)$$

The cross-covariance $C(\lambda)$ determines how strongly $X_\lambda$ predicts $V$ through linear regression: a large $\|C(\lambda)\|$ means a strong linear prediction.

When $(X_0, X_1)$ is jointly Gaussian, $v^*$ is linear:

$$v^*(x, \lambda) = A(\lambda)x + b(\lambda) \qquad (17)$$

where $A(\lambda) = C(\lambda)\Phi(\lambda)^{-1}$ and $b(\lambda) = \mathbb{E}[V] - A(\lambda)\mathbb{E}[X_\lambda]$. For non-Gaussian $p_1$, $v^*$ may have a nonlinear component.

# 4. Theoretical Analysis

Having identified $G_n^{\text{train}}(\lambda)$ as the membership signal, we now analyse its structure as a function of $\lambda$. We first identify a critical point where linear information is minimal (Section 4.1), then prove that the membership signal peaks there for Gaussian distributions (Section 4.2), and finally extend heuristically to the general case (Section 4.3). As a reminder, for readability, all proofs are deferred to Appendix A.

## 4.1. The Critical Point: Minimal Cross-Covariance

We first identify a special value of $\lambda$ where the cross-covariance $C(\lambda)$ has a minimal norm.

**Proposition 4.1** (Critical point of cross-covariance). *The squared Frobenius norm $\|C(\lambda)\|_F^2$ is a convex parabola in $\lambda$ with a unique minimum at:*

$$\lambda_F^* = \frac{\text{tr}(\Sigma_0^2) + \text{tr}(\Sigma_0 \Sigma_1)}{\text{tr}((\Sigma_0 + \Sigma_1)^2)} \qquad (18)$$

Under isotropy ($\Sigma_0 = \sigma_0^2 I$, $\Sigma_1 = \sigma_1^2 I$), this minimum has a stronger interpretation: $C(\lambda_F^*) = 0$ exactly, so the optimal linear predictor $A(\lambda) = C(\lambda)\Phi(\lambda)^{-1}$ vanishes and $X_\lambda$ carries no linear information about $V$. In the general case, minimising $\|C(\lambda)\|_F$ does not guarantee $A(\lambda)$ is minimised, since $\Phi(\lambda)^{-1}$ also varies with $\lambda$.

## 4.2. Gaussian Case: Peak at Minimal Linear Information

For isotropic Gaussian distributions, we prove that $\mathbb{E}_{\mathcal{D}^{\text{train}}}[G_n^{\text{train}}(\lambda)]$ is maximised exactly at $\lambda_F^*$.

**Theorem 4.2** (Peak location for isotropic Gaussian). *Let $X_0 \sim \mathcal{N}(0, \sigma_0^2 I_d)$ and $X_1 \sim \mathcal{N}(0, \sigma_1^2 I_d)$ be independent in $\mathbb{R}^d$. For a linear model trained by ordinary least squares on $n > 2$ samples:*

$$\mathbb{E}_{\mathcal{D}^{\text{train}}}[G_n^{\text{train}}(\lambda)] = \sigma_{\text{irr}}^2(\lambda) \cdot \frac{n-1}{n(n-2)} \qquad (19)$$

*where $\sigma_{\text{irr}}^2(\lambda) = d\big(\sigma_0^2 + \sigma_1^2 - c(\lambda)^2/\phi(\lambda)\big)$, with scalars $\phi(\lambda) = (1-\lambda)^2 \sigma_0^2 + \lambda^2 \sigma_1^2$ and $c(\lambda) = \lambda \sigma_1^2 - (1-\lambda)\sigma_0^2$.*

**Corollary 4.3** (Peak at minimal linear information). *Under the assumptions of Theorem 4.2, $\mathbb{E}_{\mathcal{D}^{\text{train}}}[G_n^{\text{train}}(\lambda)]$ is uniquely maximised at:*

$$\lambda^* = \frac{\sigma_0^2}{\sigma_0^2 + \sigma_1^2} \qquad (20)$$

*This coincides with $\lambda_F^*$ from Proposition 4.1 in the isotropic case.*

**Corollary 4.4** (Boundary behavior). *Under the assumptions of Theorem 4.2, $\mathbb{E}_{\mathcal{D}^{\text{train}}}[G_n^{\text{train}}(\lambda)]$ is minimised at $\lambda \in \{0, 1\}$.*

**Corollary 4.5** (Asymptotics behavior). *For large $n$:*

$$\mathbb{E}[G_n^{\text{train}}(\lambda)] \approx \frac{\sigma_{\text{irr}}^2(\lambda)}{n} \qquad (21)$$

## 4.3. General Case: Heuristic Extension

For non-Gaussian distributions and nonlinear models, we provide heuristic arguments that we validate empirically in Section 6.

**Decomposition of the learning target.** For general distributions, the optimal predictor may have a nonlinear component:

$$v^*(x, \lambda) = A(\lambda)x + b(\lambda) + r(x, \lambda) \qquad (22)$$

where $r(x, \lambda) \triangleq v^*(x, \lambda) - A(\lambda)x - b(\lambda)$ captures the deviation from the best linear approximation. For Gaussian distributions, $r \equiv 0$.

For a training sample $i$, the target velocity becomes:

$$v^{(i)} = \underbrace{A(\lambda)x_\lambda^{(i)} + b(\lambda)}_{\text{linear signal}} + \underbrace{r(x_\lambda^{(i)}, \lambda) + \epsilon_i(\lambda)}_{\eta_i(\lambda): \text{ nonlinear target}} \qquad (23)$$

The linear signal generalises to held-out data. The nonlinear target $\eta_i(\lambda)$ combines the population nonlinearity $r$ (which generalises) with the sample-specific residual $\epsilon_i$ (which does not).

**The competition mechanism.** From the perspective of gradient descent, the model cannot distinguish between $r(x_\lambda^{(i)}, \lambda)$ and $\epsilon_i(\lambda)$. This indistinguishability follows from their shared statistical structure:

**Proposition 4.6** (Shared statistics of $r$ and $\epsilon$). *For* $(X_0, X_1) \sim p_0 \times p_1$:

$$\mathbb{E}_{p_0 \times p_1}[r(X_\lambda, \lambda)] = 0 \quad (24)$$

$$\text{Cov}_{p_0 \times p_1}(r(X_\lambda, \lambda), X_\lambda) = 0 \quad (25)$$

$$\mathbb{E}_{p_0 \times p_1}[\epsilon(\lambda)] = 0 \quad (26)$$

$$\text{Cov}_{p_0 \times p_1}(\epsilon(\lambda), X_\lambda) = 0 \quad (27)$$

Since $\mathcal{D}_{\text{train}}$ is drawn i.i.d. from $p_0 \times p_1$, the law of large numbers implies:

$$\frac{1}{n}\sum_{i=1}^{n}\eta_i(\lambda) \xrightarrow{n \to \infty} 0, \quad \frac{1}{n}\sum_{i=1}^{n}\eta_i(\lambda)(x_\lambda^{(i)})^\top \xrightarrow{n \to \infty} 0 \quad (28)$$

A model observing only $\{(x_\lambda^{(i)}, v^{(i)})\}_{i=1}^{n}$ cannot distinguish, based on first and second-order statistics, which part of $\eta_i$ will generalise. The gradient pushes the model to explain $\eta_i = r + \epsilon_i$ jointly, inevitably fitting some of the sample-specific component $\epsilon_i$.

**Role of the linear signal.** When $\|C(\lambda)\|$ is large, the linear signal dominates, and by spectral bias (Rahaman et al., 2019), it is learnt first. The nonlinear target $\eta_i$ contributes little to the loss, keeping $G_n^{\text{train}}(\lambda)$ low.

Near $\lambda_F^*$, where $\|C(\lambda)\|$ is minimised (Proposition 4.1), the linear signal vanishes ($A(\lambda) \approx 0$). The model must explain the entirety of $\eta_i$ using nonlinear features. Competition between learning $r$ and fitting $\epsilon_i$ is maximal, and $G_n^{\text{train}}(\lambda)$ peaks.

### 4.4. Why Standard Metrics Miss the Membership Signal

The train-test gap $\Delta(\lambda) \approx 2G_n^{\text{train}}(\lambda)$ provides a membership signal at each $\lambda$. Yet standard training protocols fail to detect it due to two masking mechanisms.

**Spatial averaging.** Standard training monitors losses averaged over $\lambda \sim p(\lambda)$:

$$L_{\text{global}} = \mathbb{E}_{\lambda \sim p(\lambda)}[L(\lambda)] \quad (29)$$

If $G_n^{\text{train}}(\lambda)$ concentrates near $\lambda_F^*$ while $p(\lambda)$ spreads over $[0, 1]$, the signal is diluted.

**Temporal compensation.** On the training data, the loss decomposes as $L_{\text{train}}(\lambda) = E_n^{\text{train}}(\lambda) + \hat{\sigma}_n^2(\lambda) - 2G_n(\lambda)$. As training progresses, $E_n^{\text{train}}(\lambda)$ decreases while $G_n(\lambda)$ increases as the model fits training-specific residuals; both effects reduce $L_{\text{train}}$, making them indistinguishable.

On validation data, under Assumption 3.2, $E^{\text{test}}(\lambda)$ decreases in tandem while $G^{\text{test}}(\lambda) \approx 0$, so validation loss also decreases. The membership signal thus accumulates

while both losses improve, leaving the model vulnerable to membership inference at early stopping despite no visible overfitting.

### 4.5. Why the Assumptions Hold in Practice

The closed-form prediction $\lambda_F^*$ relies on Gaussian isotropic assumptions. We argue that these are reasonable approximations for latent diffusion models. We discuss here why these are reasonable approximations for latent diffusion models, how their validity can be characterised empirically, and what alternatives exist when they fail.

**Approximate Gaussianity.** Latent spaces are designed with constrained statistics: VAEs regularise toward a Gaussian prior via KL divergence (Kingma & Welling, 2014), while encoders like Music2Latent (Pasini et al., 2024) bind activations via tanh. By the maximum entropy principle (Jaynes, 1957), bounded latent spaces with fixed covariance tend toward Gaussian distributions.

**Approximate isotropy.** KL-regularised VAEs explicitly penalise deviation from $\mathcal{N}(0, I)$ (Kingma & Welling, 2014). For other encoders, architectural choices produce similar effects: batch normalisation (Ioffe & Szegedy, 2015) standardises activations, and symmetric bounded activations like tanh discourage correlations. More generally, independent per-dimension processing tends toward approximately diagonal covariance.

**Dominant linear structure.** Even when the nonlinear residual $r$ is non-zero, neural networks learn low-frequency (linear) components first due to spectral bias (Rahaman et al., 2019). Xu et al. (2019) formalises this as the Frequency Principle: networks fit target functions from low to high frequencies during training. The location where $A(\lambda)$ vanishes determines where the model must rely on higher-order structure.

**Continuity of $\lambda_F^*$.** The formula for $\lambda_F^*$ depends continuously on $\Sigma_0$ and $\Sigma_1$. Small deviations from exact Gaussianity or isotropy produce correspondingly small deviations in the peak location, suggesting robustness to moderate violations.

## 5. Experimental Protocol

Having established that the train-test gap follows a bell-shaped curve over $\lambda$ peaking at $\lambda_F^*$, we now design a protocol to validate these predictions and demonstrate their exploitation for membership inference.

## 5.1. Detection Protocol

Given a trained model $v_\theta$ and a sample $x_1$, we measure the reconstruction quality at each $\lambda$:

1. **Interpolate**: Sample $x_0 \sim p_0 = \mathcal{N}(0, \Sigma_0)$, compute $x_\lambda = (1 - \lambda)x_0 + \lambda x_1$

2. **Predict**: Compute $v_\theta(x_\lambda, \lambda)$

3. **Reconstruct**: $\hat{x}_1 = x_\lambda + (1 - \lambda)v_\theta(x_\lambda, \lambda)$

4. **Measure**: $\text{MSE}(\lambda) = \|x_1 - \hat{x}_1\|^2$

For each data point $x_1$, we sample $K = 100$ different noise samples $x_0$ and average the MSE over them. We vary $\lambda \in \{0, 0.1, \ldots, 1.0\}$. The procedure is depicted in Figure 1.

## 5.2. Experimental Setup

We first establish a baseline configuration on audio, then systematically vary each component to test robustness. In the following, we present our baseline, and full details on all datasets, encoders, and architectures are in Appendix B.

**Baseline configuration.** Our primary experiments use MAESTRO v3 (Hawthorne et al., 2019), a dataset of $\sim$200 hours of classical piano, where the official split ensures no composition appears in multiple subsets (satisfying Assumption 3.3). Audio is encoded via Music2Latent (Pasini et al., 2024), a pretrained autoencoder mapping to 64-channel latents at 10 Hz. We train a Transformer (410M parameters) adapted from DiT (Peebles & Xie, 2023) with AdamW (lr $10^{-4}$, batch size 256), log normal $\lambda$-sampling (Esser et al., 2024), and early stopping at the validation plateau.

**Evaluation.** We compute reconstruction MSE on 5,000 training and 5,000 held-out samples, using $K = 100$ noise realisations per sample. The reconstruction error satisfies $\text{MSE}(\lambda) = (1 - \lambda)^2\|v_\theta - v\|^2$. Early stopping ensures Assumption 3.2 while $G_n(\lambda)$ accumulates undetected. We normalise to remove the $(1 - \lambda)^2$ factor:

$$\Delta_{\text{norm}}(\lambda) = \frac{\text{MSE}^{\text{test}}(\lambda) - \text{MSE}^{\text{train}}(\lambda)}{\text{MSE}^{\text{test}}(\lambda) + \text{MSE}^{\text{train}}(\lambda)} \quad (30)$$

This quantity is proportional to $G_n(\lambda)$ and peaks at $\lambda_F^*$, being positive when the model reconstructs training samples better than held-out ones.

**Ablations.** To validate the robustness of our findings, we vary the configuration along several axes: (1) the data distribution $\Sigma_1$ (datasets of different diversity: FMA Large (Defferrard et al., 2017a), MTG Jamendo (Bogdanov et al., 2019)), (2) the noise distribution $\Sigma_0$ (scaling the variance by

$\times 4$ and $\times 1/4$), (3) the latent space (Music2Latent vs Stable Audio VAE (Evans et al., 2025)), (4) the modality (audio vs images using CelebA datasets (Liu et al., 2015)), (5) the architecture (Transformer vs UNet), (6) the model capacity (from 410M to 140M and 880M parameters), and (7) the sampling scheduler (uniform vs log-normal). Results are in Section 6.2. Full details on datasets, architectures, and configurations are provided in Appendix B.

*Table 1.* Gaussianity and isotropy of latent representations. $\overline{|\gamma|}$: mean absolute skewness (0 for symmetric); $\overline{|\kappa|}$: mean excess kurtosis (0 for Gaussian); $\overline{|\rho|}$: mean absolute inter-dimension correlation (0 for independent); $\|\Sigma - I\|_F/d$: normalised deviation from isotropic unit variance.

| DATASET | LATENT SPACE | $\overline{|\gamma|}$ | $\overline{|\kappa|}$ | $\overline{|\rho|}$ | $\|\Sigma - I\|_F/d$ |
|---|---|---|---|---|---|
| MAESTRO V3 | MUSIC2LATENT | 0.18 | 0.22 | 0.23 | 0.14 |
| MTG-JAMENDO | MUSIC2LATENT | 0.07 | 0.16 | 0.17 | 0.13 |
| FMA LARGE | MUSIC2LATENT | 0.08 | 0.23 | 0.16 | 0.12 |
| MAESTRO V3 | STABLE AUDIO VAE | 0.08 | 0.10 | 0.16 | 0.08 |
| CELEBA | STABLE DIFFUSION VAE | 0.12 | **0.71** | **0.61** | **0.40** |

## 6. Results

### 6.1. Validating Theoretical Predictions

Before presenting our main results, we verify that both our latent spaces and the model architecture satisfy the assumptions underlying Theorem 4.2.

**Gaussianity of latent representations.** Theorem 4.2 proves that the membership signal peaks at $\lambda_F^*$ under Gaussian isotropic assumptions. Table 1 reports skewness, excess kurtosis, and covariance isotropy for each configuration. As we see, the latent spaces of all our audio configurations exhibit low skewness, kurtosis, and weak inter-dimension correlations, satisfying our Gaussian isotropic assumptions. In contrast, the latent space of our image configuration (CelebA with Stable Diffusion VAE) does not satisfy the required assumptions. While its skewness remains acceptable ($\overline{|\gamma|} = 0.12$), its kurtosis ($\overline{|\kappa|} = 0.71$) indicates heavy-tailed marginals, and the correlations are strong ($\overline{|\rho|} = 0.61$).

**Mechanistic assumptions: competition between linear and nonlinear features.** The closed-form prediction $\lambda_F^*$ and the heuristic argument of Section 4.3 rely on the model leveraging nonlinear features near $\lambda_F^*$, where linear prediction becomes impossible. We test this directly by comparing the trained Transformer to a linear OLS predictor fitted on the same task. Figure 2 reports the ratio of their test losses as a function of $\lambda$. As predicted, the ratio is close to 1 at the boundaries $\lambda \in \{0, 1\}$, where linear prediction suffices, and peaks near where the membership signal is maximum, i.e., where the Transformer's nonlinear capacity provides

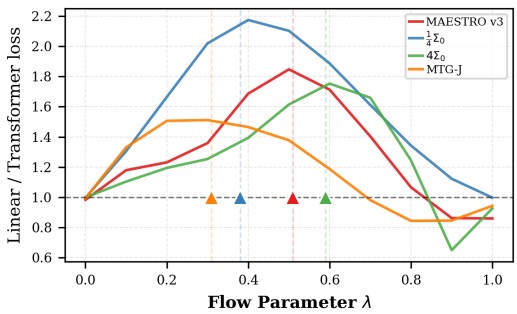

Figure 2. Ratio of Transformer to OLS test loss as a function of $\lambda$, across configurations. The ratio is consistently maximal where the membership signal peaks.

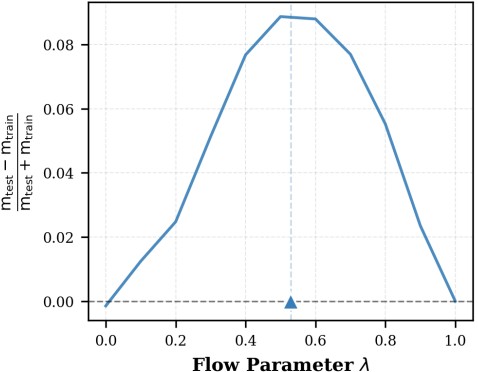

Figure 3. Normalised train-test gap $\Delta_{\mathrm{norm}}(\lambda)$ on MAESTRO. The curve exhibits the predicted bell shape with peak near $\lambda_F^*$ (dashed line).

the largest gain. The pattern holds across multiple configurations.

**Bell-shaped gap curve.** Figure 3 displays the normalised train-test gap $\Delta_{\mathrm{norm}}(\lambda)$ on MAESTRO v3. As predicted, the gap exhibits a bell-shaped pattern: minimal at boundaries ($\lambda \in \{0, 1\}$) and maximal at intermediate values, confirming Corollary 4.4. This bell shape is universal; it appears in all configurations we tested, regardless of dataset, architecture, latent space, or modality (Section 6.2). Extended analysis including additional statistics is provided in Appendix C.

**Peak location.** For configurations satisfying the Gaussian isotropic assumptions, the observed peak $\lambda_{\mathrm{obs}}$ matches the theoretical prediction $\lambda_F^*$ from Proposition 4.1 within grid resolution. On MAESTRO v3 with Music2Latent, $\lambda_{\mathrm{obs}} \in [0.5, 0.6]$ versus $\lambda_F^* = 0.52$. This agreement holds across all audio configurations (Table 2).

**Temporal evolution.** A central claim is that the membership signal differs from classical overfitting. Figure 4

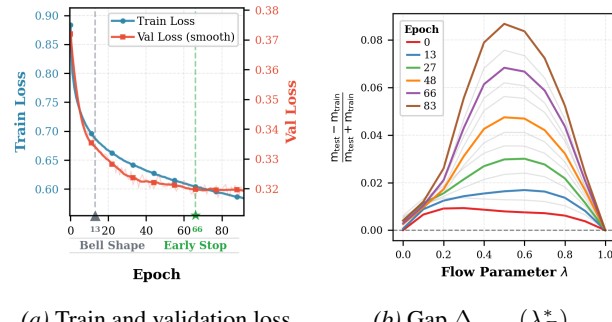

*(a)* Train and validation loss    *(b)* Gap $\Delta_{\mathrm{norm}}(\lambda_F^*)$

Figure 4. Temporal evolution on MAESTRO. (a) Validation loss decreases until early stopping (dashed). (b) Train-test gap grows throughout training.

provides direct evidence. Validation loss decreases steadily until early stopping (Figure 4a), suggesting healthy learning according to standard diagnostics. Yet the gap $\Delta_{\mathrm{norm}}(\lambda_F^*)$ grows from the first epochs (Figure 4b), long before validation plateaus. By early stopping, a significant gap has accumulated, which is invisible to standard metrics but exploitable for membership inference.

### 6.2. Ablation Study

Table 2 summarises all configurations tested. The bell-shaped curve appears in every case; the peak prediction $\lambda_F^*$ matches when the Gaussian isotropic assumptions hold.

**(1) Data distribution ($\Sigma_1$).** Figure 5 shows bell curves for three audio datasets with varying diversity and covariance $\Sigma_1$, testing Proposition 4.1: each yields a different predicted $\lambda_F^*$, and observed peaks match in all cases (Table 2, rows 1). Peak magnitude varies with dataset size; MAESTRO v3 (smallest) shows the strongest signal, while FMA Large (largest) shows the weakest, consistent with the $\sim 1/n$ scaling of the membership signal predicted by Corollary A.7.

**(2) Noise distribution ($\Sigma_0$).** Figure 5 also shows the effect of scaling the noise variance while fixing $\Sigma_1$ using the Maestrov3 dataset, directly testing Proposition 4.1: increasing $\sigma_0^2$ shifts $\lambda_F^*$ rightward as predicted (Table 2, row 2). For $\Sigma_0 \times 4$, the predicted $\lambda_F^* = 0.59$ falls just below the observed interval $[0.6, 0.7]$, which we consider a match within grid resolution.

**(3) Latent space.** Replacing Music2Latent with Stable Audio VAE yields a different predicted $\lambda_F^*$ (0.50 vs. 0.52), as expected since the two encoders induce different covariances $\Sigma_1$. The observed peak matches the prediction in both cases (Table 2, rows 3; Figure 6).

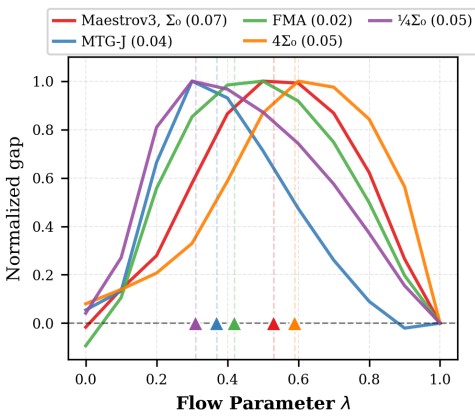

*Figure 5.* Ablations (1)–(2): Effect of data distribution $\Sigma_1$ and noise distribution $\Sigma_0$. Values are normalise for better visualisation, Value between parentheses are raw values. Dashed lines indicate predicted $\lambda_F^*$ values. Trained with Maestrov3 dataset with Music2Latent latent space and Transformer architecture

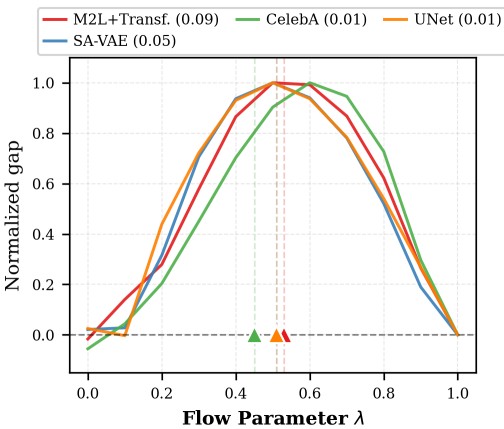

*Figure 6.* Ablations (3)–(5): Effect of latent space encoder, model architecture, and modality. Values are normalise for better visualisation, Value between parentheses are raw values. The bell shape persists across all configurations; peak prediction fails only when Gaussian isotropic assumptions are violated (CelebA).

**(4) Modality: limits of $\lambda_F$ prediction.** On CelebA with SD VAE, the bell-shaped curve persists (Figure 6), confirming that the phenomenon extends beyond audio. However, the observed peak ($\lambda_{\text{obs}} \in [0.6, 0.7]$) deviates from the prediction ($\lambda_F^* = 0.45$; Table 2).The high kurtosis and correlation values (Table 1) violate Theorem 4.2's requirement, suggesting why peak prediction fails for this configuration. A discussion about the analysis of these failure modes, along with an exploration of the possibility of relaxation, is provided in Appendix D.

**(5) Architecture.** Replacing the Transformer with a UNet preserves both the bell-shaped structure and the peak location $\lambda_F^*$ (Table 2, ablation (4); Figure 6). However, the peak magnitude drops substantially (from 0.09 to 0.01), consistent with the UNet producing notably lower-quality generations than the Transformer.

**(6) Model capacity.** Varying the Transformer size from 140M to 880M parameters leaves the peak location unchanged across all configurations (Table 2, rows 6; Figure 7), while peak magnitude increases consistently with model size. Larger models fit training-specific residuals more accurately, amplifying the membership signal without shifting its location.

**(7) $\lambda$-sampling scheduler.** Replacing the log-normal scheduler with a uniform one preserves both the bell-shaped curve and the peak location while reducing the peak magnitude (Table 2, rows 7; Figure 7). This attenuation is consistent with the log-normal scheduler concentrating training near $\lambda \approx 0.5$, which coincides with $\lambda_F^*$ and thereby amplifies the membership signal.

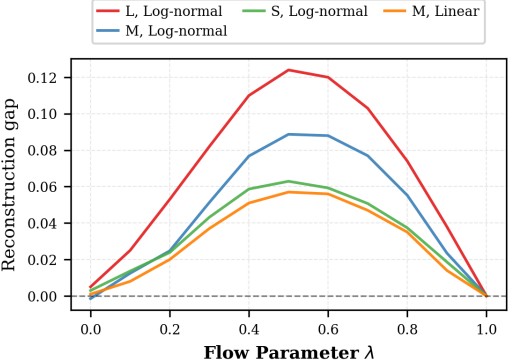

*Figure 7.* Ablations (6)–(7): Effect of model capacity and $\lambda$-sampling scheduler. The peak location remains unchanged across all configurations; only the magnitude varies. Sizes: S is 140M parameters, M is 410M parameters and L is 880M parameters

### 6.3. What holds universally vs. what requires our assumptions

Across all configurations, the bell-shaped structure, its boundary behaviour, its temporal accumulation, and the linear/nonlinear competition mechanism hold universally, including for CelebA, where our Gaussian isotropic assumptions are violated (ablations 1–7). Within this universal structure, peak *location* is governed solely by data geometry $(\Sigma_0, \Sigma_1)$: dataset, noise scale, and encoder shift predictably per Proposition 4.1 (ablations 1–3), while architecture, capacity, and scheduler do not (ablations 4, 6, 7). Peak *magnitude*, by contrast, reflects model and training choices: larger models and log-normal scheduling amplify the signal without moving its location (ablations 6–7).

*Table 2.* Ablation study summary. All configurations exhibit the bell-shaped curve. For ablations (1)–(4), the predicted peak $\lambda_F^*$ matches $\lambda_{\text{obs}}$ when Gaussian isotropic assumptions hold. For ablations (5)–(7), the peak location is unchanged while magnitude varies. †: assumptions violated (see Table 1).

| | ABLATION | CONFIGURATION | $\lambda_F^*$ | $\lambda_{\text{obs}}$ | MATCH |
|---|---|---|---|---|---|
| (1) | DATA ($\Sigma_1$) | MAESTRO V3 | 0.52 | 0.5–0.6 | ✓ |
| | | MTG-JAMENDO | 0.37 | 0.3–0.4 | ✓ |
| | | FMA LARGE | 0.42 | 0.4–0.5 | ✓ |
| (2) | NOISE ($\Sigma_0$) | $\Sigma_0 \times 0.25$ | 0.31 | 0.3–0.4 | ✓ |
| | | $\Sigma_0 \times 1$ | 0.52 | 0.5–0.6 | ✓ |
| | | $\Sigma_0 \times 4$ | 0.59 | 0.6–0.7 | ✓ |
| (3) | LATENT SPACE | MUSIC2LATENT | 0.52 | 0.5–0.6 | ✓ |
| | | STABLE AUDIO VAE | 0.50 | 0.5–0.6 | ✓ |
| (4) | MODALITY† | CELEBA (SD VAE) | 0.45 | 0.6–0.7 | ✗ |

| | ABLATION | CONFIGURATION | PEAK MAGNITUDE |
|---|---|---|---|
| (5) | ARCHITECTURE | TRANSFORMER | 0.09 |
| | | UNET | 0.01 |
| (6) | MODEL CAPACITY | 140M | 0.06 |
| | | 410M | 0.09 |
| | | 880M | 0.12 |
| (7) | SCHEDULER | LOG-NORMAL | 0.09 |
| | | UNIFORM | 0.06 |

## 7. Implications for Membership Inference

Our analysis reveals that the reconstruction error follows a predictable bell-shaped profile across $\lambda$, with a computable peak location and a vanishing signal at the boundaries. As a proof of concept, we demonstrate that this structured gap is exploitable for MIA.

**Membership Inference Attack.** Given a query sample $x_1$, we compute the reconstruction MSE at each $\lambda \in \{0, 0.1, \dots, 1.0\}$ using $K = 100$ noise samples, yielding an 11-dimensional feature vector that captures the full $\lambda$-resolved profile. We then train a simple MLP classifier on these features to predict member/non-member, requiring only forward passes through the trained model (no gradient computation or weight access), making the attack lightweight and practical. The $\lambda$-resolved profile provides a richer signal than any single evaluation point: it encodes the full shape of the bell curve, whose amplitude and location are characteristic of training membership. Using a single reconstruction error at $\lambda = \lambda^*$, equivalent to ignoring the $\lambda$-resolved structure (i.e. Naive Attack), achieves only a 0.67 AUC score. Consistent with our theory, the naive baseline (i.e., using a single reconstruction error) peaks at $\lambda^*$, confirming that the membership signal concentrates there as predicted. Adapting SecMI (Duan et al., 2023) and PIA (Kong et al., 2023) to Rectified Flows yields AUC scores of 0.72 and 0.83, respectively. Our method achieves 0.91 AUC on MAESTRO V3, demonstrating that the theoretical signal translates to practical risks. Results on additional datasets, in Appendix E, remain positive across all configurations, with AUC scores decreasing consistently with the amplitude of the bell-shaped gap observed for each dataset.

## 8. Discussion

**Limitations.** The closed-form peak prediction $\lambda_F^*$ requires near-Gaussian isotropic latents; on CelebA with SD VAE, the peak location deviates, though the bell shape persists, confirming it is a universal property of Rectified Flow training independent of our distributional assumptions. Our theory also assumes independent coupling ($X_0 \perp\!\!\!\perp X_1$), excluding the reflow procedure; preliminary experiments (Appendix F) suggest the bell shape persists under one reflow step, but with substantially attenuated magnitude, indicating reflow may offer a natural mitigation as a byproduct of its trajectory-straightening objective. The MIA we developed is a proof of concept under a white-box setting; stronger threat models, such as black-box or label-only access, remain to be explored. We also study unconditional generation exclusively, while deployed systems condition on text prompts; conditioning modifies the effective distribution, altering $\Sigma_1$ and hence $\lambda_F^*$. Finally, our experiments scale up to 880M parameters; model capacity amplifies the signal (ablation 6), while dataset size attenuates it (ablation 1), and their interaction at the scale of deployed systems such as FLUX or SD3 remains an open empirical question.

**Implications.** Since $\lambda_F^*$ is architecture-independent (ablations 4–7), the peak can be located empirically on a small proxy model and transferred to larger target models without retraining. This structural knowledge also opens the door to targeted defences: rather than regularising uniformly across the interpolation path, one could concentrate privacy-preserving mechanisms near $\lambda_F^*$, where the membership signal is maximal. Beyond security, our analysis connects to training efficiency: the peak $\lambda^*$ corresponds to where prediction is hardest, as $x_\lambda$ contains balanced contributions from noise and data. Esser et al. (2024) found empirically that concentrating $p(\lambda)$ near 0.5 improves SD3; our theory provides a principled explanation and suggests that adapting $p(\lambda)$ to dataset-specific $\lambda^*$ could further accelerate convergence. Conversely, schedulers concentrated near $\lambda_F^*$ also amplify membership leakage, revealing a fundamental trade-off between training efficiency and privacy.

## 9. Conclusion

We showed that Rectified Flows encode membership signals in a structured, predictable way: it follows a universal bell-shaped curve over $\lambda$, peaks at a location governed by data geometry, and accumulates silently while standard diagnostics see nothing. This structure translates into practical risk, as a simple MIA exploiting it consistently outperforms baselines adapted from the diffusion literature.

## Impact Statement

This work aims to improve the theoretical understanding of Rectified Flows and the information they retain about their training data. We hope it provides useful tools for practitioners and researchers working on generative models.

## Acknowledgements

We thank the anonymous reviewers for their thorough and insightful reviews. We are also grateful to Manuel Moussalam and Romain Hennequin from Deezer for their careful reading of the mathematical derivations and valuable feedback during the preparation of this manuscript. This work was supported by the computational resources provided by LTCI, Télécom Paris, Institut Polytechnique de Paris, Palaiseau, France.

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

# A. Proofs

## A.1. Proof of Proposition 3.1

*Proof.* Conditioning on $\mathcal{D}^{\text{train}}$ fixes the trained model $v_\theta$ as a deterministic function. Define $g : \mathbb{R}^d \to \mathbb{R}^d$ by $g(x) = v_\theta(x, \lambda) - v^*(x, \lambda)$.

Each test sample $(\tilde{x}_0^{(j)}, \tilde{x}_1^{(j)})$ is drawn i.i.d. from $p_0 \times p_1$, independently of $\mathcal{D}^{\text{train}}$. By the orthogonality property (4):

$$\mathbb{E}_{p_0 \times p_1}[\langle g(X_\lambda), V - v^*(X_\lambda, \lambda)\rangle] = 0 \tag{31}$$

Therefore, for each test sample $j$:

$$\mathbb{E}\big[\langle v_\theta(\tilde{x}_\lambda^{(j)}, \lambda) - v^*(\tilde{x}_\lambda^{(j)}, \lambda), \tilde{\epsilon}_j(\lambda)\rangle \mid \mathcal{D}^{\text{train}}\big] = 0 \tag{32}$$

By linearity: $\mathbb{E}_{\mathcal{D}^{\text{test}}}[G_m^{\text{test}}(\lambda) \mid \mathcal{D}^{\text{train}}] = 0$.

On training data, this argument does not apply: both $v_\theta$ and $\{\epsilon_i(\lambda)\}_{i=1}^n$ depend on $\mathcal{D}^{\text{train}}$, so $g$ is not independent of the residuals. $\qquad\square$

## A.2. Proof of Proposition 4.1: Critical point of cross-covariance

*Proof.* Expand $\|C(\lambda)\|_F^2 = \|\lambda\Sigma_1 - (1-\lambda)\Sigma_0\|_F^2$:

$$\begin{aligned} \|C(\lambda)\|_F^2 = \lambda^2 \text{tr}(\Sigma_1^2) + (1-\lambda)^2\text{tr}(\Sigma_0^2) \\ - 2\lambda(1-\lambda)\text{tr}(\Sigma_0\Sigma_1) \end{aligned} \tag{33}$$

Rearranging:

$$\begin{aligned} \|C(\lambda)\|_F^2 = \lambda^2\text{tr}((\Sigma_0 + \Sigma_1)^2) \\ - 2\lambda\big(\text{tr}(\Sigma_0^2) + \text{tr}(\Sigma_0\Sigma_1)\big) + \text{tr}(\Sigma_0^2) \end{aligned} \tag{34}$$

This is a convex parabola with a positive leading coefficient. The minimum is at $\lambda_F^*$. $\qquad\square$

## A.3. Proof of Theorem 4.2: Isotropic Gaussian Case

We provide a complete analysis of the expected train-test gap in the isotropic Gaussian setting.

### A.3.1. SETUP

**Assumption A.1** (Isotropic Gaussian). Let $X_0 \sim \mathcal{N}(0, \sigma_0^2 I_d)$ and $X_1 \sim \mathcal{N}(0, \sigma_1^2 I_d)$ be independent, with $\sigma_0, \sigma_1 > 0$.

Define $X_\lambda = (1-\lambda)X_0 + \lambda X_1$ and $V = X_1 - X_0$. In the isotropic case, the covariance matrices from Section 3 reduce to scalars times the identity:

$$\begin{aligned} \Phi(\lambda) = \phi(\lambda)I_d && \text{where} && \phi(\lambda) = (1-\lambda)^2\sigma_0^2 + \lambda^2\sigma_1^2 \end{aligned} \tag{35}$$

$$\begin{aligned} C(\lambda) = c(\lambda)I_d && \text{where} && c(\lambda) = \lambda\sigma_1^2 - (1-\lambda)\sigma_0^2 \end{aligned} \tag{36}$$

Since the covariances are isotropic, all coordinates are independent and identically distributed. We analyse a single coordinate $j$, then sum over $d$ coordinates.

For coordinate $j$:

$$X_{\lambda,j} \sim \mathcal{N}(0, \phi(\lambda)) \tag{37}$$

$$V_j \sim \mathcal{N}(0, \sigma_0^2 + \sigma_1^2) \tag{38}$$

$$\text{Cov}(V_j, X_{\lambda,j}) = c(\lambda) \tag{39}$$

### A.3.2. OPTIMAL PREDICTOR

Since $(X_{\lambda,j}, V_j)$ is jointly Gaussian with zero means, the conditional expectation is linear:

$$v_j^*(x, \lambda) = \mathbb{E}[V_j \mid X_{\lambda,j} = x] = a(\lambda) \cdot x \tag{40}$$

where:

$$a(\lambda) = \frac{\text{Cov}(V_j, X_{\lambda,j})}{\text{Var}(X_{\lambda,j})} = \frac{c(\lambda)}{\phi(\lambda)} \tag{41}$$

### A.3.3. IRREDUCIBLE VARIANCE

**Lemma A.2** (Irreducible variance). *Under Assumption A.1:*

$$\sigma_{\text{irr}}^2(\lambda) = d\left(\sigma_0^2 + \sigma_1^2 - \frac{c(\lambda)^2}{\phi(\lambda)}\right) \tag{42}$$

*Proof.* The residual for coordinate $j$ is $\epsilon_j(\lambda) = V_j - a(\lambda)X_{\lambda,j}$. Its variance is:

$$\text{Var}(\epsilon_j(\lambda)) = \text{Var}(V_j) - 2a(\lambda)\text{Cov}(V_j, X_{\lambda,j}) + a(\lambda)^2\text{Var}(X_{\lambda,j}) \tag{43}$$

$$= (\sigma_0^2 + \sigma_1^2) - 2\frac{c(\lambda)}{\phi(\lambda)} \cdot c(\lambda) + \frac{c(\lambda)^2}{\phi(\lambda)^2} \cdot \phi(\lambda) \tag{44}$$

$$= \sigma_0^2 + \sigma_1^2 - \frac{c(\lambda)^2}{\phi(\lambda)} \triangleq \sigma_\epsilon^2(\lambda) \tag{45}$$

Summing over $d$ independent coordinates gives $\sigma_{\text{irr}}^2(\lambda) = d \cdot \sigma_\epsilon^2(\lambda)$. $\square$

### A.3.4. OLS ESTIMATOR

For each coordinate $j$, we have $n$ i.i.d. samples $(x_{\lambda,j}^{(i)}, v_j^{(i)})_{i=1}^n$ with the model:

$$v_j^{(i)} = a(\lambda)x_{\lambda,j}^{(i)} + \epsilon_j^{(i)}(\lambda) \tag{46}$$

where $\epsilon_j^{(i)}(\lambda) \sim \mathcal{N}(0, \sigma_\epsilon^2(\lambda))$ and $\epsilon_j^{(i)}(\lambda) \perp x_{\lambda,j}^{(i)}$ (by Gaussianity).

This is a univariate linear regression without intercept. The OLS estimator is:

$$\hat{a} = \frac{\sum_{i=1}^n v_j^{(i)} x_{\lambda,j}^{(i)}}{\sum_{i=1}^n (x_{\lambda,j}^{(i)})^2} = \frac{\sum_{i=1}^n v_j^{(i)} x_{\lambda,j}^{(i)}}{S} \tag{47}$$

where $S \triangleq \sum_{i=1}^n (x_{\lambda,j}^{(i)})^2$.

Substituting $v_j^{(i)} = a(\lambda)x_{\lambda,j}^{(i)} + \epsilon_j^{(i)}(\lambda)$:

$$\hat{a} = a(\lambda) + \frac{\sum_i \epsilon_j^{(i)}(\lambda)x_{\lambda,j}^{(i)}}{S} \tag{48}$$

### A.3.5. DISTRIBUTION OF $S$

Since $x_{\lambda,j}^{(i)} \sim \mathcal{N}(0, \phi(\lambda))$, we have $x_{\lambda,j}^{(i)}/\sqrt{\phi(\lambda)} \sim \mathcal{N}(0, 1)$. Therefore:

$$\frac{S}{\phi(\lambda)} = \sum_{i=1}^n \left(\frac{x_{\lambda,j}^{(i)}}{\sqrt{\phi(\lambda)}}\right)^2 \sim \chi_n^2 \tag{49}$$

For $Y \sim \chi_n^2$ with $n > 2$, a standard result gives $\mathbb{E}[1/Y] = 1/(n-2)$. Hence:

$$\mathbb{E}\left[\frac{1}{S}\right] = \frac{1}{\phi(\lambda)} \cdot \mathbb{E}\left[\frac{1}{\chi_n^2}\right] = \frac{1}{\phi(\lambda)(n-2)} \tag{50}$$

### A.3.6. EXPECTED TRAINING LOSS

**Lemma A.3** (Expected training loss per coordinate). *For a single coordinate $j$:*

$$\mathbb{E}[L_{\text{train},j}(\lambda)] = \sigma_\epsilon^2(\lambda) \cdot \frac{n-1}{n} \tag{51}$$

*Proof.* This is a standard result for OLS regression. For a model with $p$ parameters, the expected residual sum of squares satisfies:

$$\mathbb{E}\left[\sum_{i=1}^n (v_j^{(i)} - \hat{a} x_{\lambda,j}^{(i)})^2\right] = (n-p)\sigma_\epsilon^2(\lambda) \tag{52}$$

Here $p = 1$ (single parameter, no intercept), so:

$$\mathbb{E}[n \cdot L_{\text{train},j}(\lambda)] = (n-1)\sigma_\epsilon^2(\lambda) \tag{53}$$

which gives $\mathbb{E}[L_{\text{train},j}(\lambda)] = \sigma_\epsilon^2(\lambda) \cdot \frac{n-1}{n}$. $\qquad\square$

### A.3.7. EXPECTED TEST LOSS

**Lemma A.4** (Expected test loss per coordinate). *For a single coordinate $j$:*

$$\mathbb{E}[L_{\text{test},j}(\lambda)] = \sigma_\epsilon^2(\lambda) \cdot \frac{n-1}{n-2} \tag{54}$$

*Proof.* For a new test point $(x_{\lambda,j}^{\text{new}}, v_j^{\text{new}})$ independent of $\mathcal{D}^{\text{train}}$:

$$v_j^{\text{new}} = a(\lambda)x_{\lambda,j}^{\text{new}} + \epsilon_j^{\text{new}}(\lambda) \tag{55}$$

The test loss (conditional on $\mathcal{D}^{\text{train}}$) is:

$$L_{\text{test},j}(\lambda) = \mathbb{E}_{\text{new}}[(v_j^{\text{new}} - \hat{a}x_{\lambda,j}^{\text{new}})^2 \mid \mathcal{D}^{\text{train}}] \tag{56}$$

$$= \mathbb{E}_{\text{new}}[((a(\lambda) - \hat{a})x_{\lambda,j}^{\text{new}} + \epsilon_j^{\text{new}}(\lambda))^2 \mid \mathcal{D}^{\text{train}}] \tag{57}$$

Since $x_{\lambda,j}^{\text{new}} \perp \epsilon_j^{\text{new}}(\lambda)$ and both are centred:

$$L_{\text{test},j}(\lambda) = (\hat{a} - a(\lambda))^2 \cdot \phi(\lambda) + \sigma_\epsilon^2(\lambda) \tag{58}$$

Taking expectations over $\mathcal{D}^{\text{train}}$:

$$\mathbb{E}[L_{\text{test},j}(\lambda)] = \phi(\lambda) \cdot \mathbb{E}[(\hat{a} - a(\lambda))^2] + \sigma_\epsilon^2(\lambda) \tag{59}$$

We now compute $\mathbb{E}[(\hat{a} - a(\lambda))^2]$. Conditionally on $(x_{\lambda,j}^{(i)})_{i=1}^n$, the numerator $\sum_i \epsilon_j^{(i)}(\lambda)x_{\lambda,j}^{(i)}$ is Gaussian with mean 0 and variance:

$$\text{Var}\left(\sum_i \epsilon_j^{(i)}(\lambda)x_{\lambda,j}^{(i)} \mid X\right) = \sum_i (x_{\lambda,j}^{(i)})^2 \cdot \sigma_\epsilon^2(\lambda) = S \cdot \sigma_\epsilon^2(\lambda) \tag{60}$$

Therefore:

$$\mathbb{E}[(\hat{a} - a(\lambda))^2 \mid X] = \frac{\sigma_\epsilon^2(\lambda) \cdot S}{S^2} = \frac{\sigma_\epsilon^2(\lambda)}{S} \tag{61}$$

Taking expectations over $X$ and using (50):

$$\mathbb{E}[(\hat{a} - a(\lambda))^2] = \sigma_\epsilon^2(\lambda) \cdot \mathbb{E}\left[\frac{1}{S}\right] = \frac{\sigma_\epsilon^2(\lambda)}{\phi(\lambda)(n-2)} \tag{62}$$

Substituting into (59):

$$\mathbb{E}[L_{\text{test},j}(\lambda)] = \phi(\lambda) \cdot \frac{\sigma_\epsilon^2(\lambda)}{\phi(\lambda)(n-2)} + \sigma_\epsilon^2(\lambda) = \sigma_\epsilon^2(\lambda) \cdot \frac{n-1}{n-2} \tag{63}$$

$$\square$$

A.3.8. FROM GAP TO $G_n^{\text{train}}$

**Lemma A.5** (Expected gap per coordinate). *For a single coordinate $j$:*

$$\mathbb{E}[\Delta_j(\lambda)] \triangleq \mathbb{E}[L_{\text{test},j}(\lambda)] - \mathbb{E}[L_{\text{train},j}(\lambda)] = \sigma_\epsilon^2(\lambda) \cdot \frac{2(n-1)}{n(n-2)} \tag{64}$$

*Proof.*

$$\mathbb{E}[\Delta_j(\lambda)] = \sigma_\epsilon^2(\lambda) \cdot \frac{n-1}{n-2} - \sigma_\epsilon^2(\lambda) \cdot \frac{n-1}{n} \tag{65}$$

$$= \sigma_\epsilon^2(\lambda)(n-1)\left(\frac{1}{n-2} - \frac{1}{n}\right) \tag{66}$$

$$= \sigma_\epsilon^2(\lambda)(n-1) \cdot \frac{2}{n(n-2)} = \sigma_\epsilon^2(\lambda) \cdot \frac{2(n-1)}{n(n-2)} \tag{67}$$

$\square$

We now connect this gap to $G_n^{\text{train}}(\lambda)$. From the loss decomposition (9) in Section 3.3:

$$L^{\text{train}}(\lambda) = E_n^{\text{train}}(\lambda) + \hat{\sigma}_n^2(\lambda) - 2G_n^{\text{train}}(\lambda) \tag{68}$$

For OLS on Gaussian data, Assumptions 3.2 and 3.3 hold in expectation:

- The OLS estimator is unbiased, so $\mathbb{E}[E_n^{\text{train}}(\lambda)] = \mathbb{E}[E^{\text{test}}(\lambda)]$
- By the law of large numbers, $\mathbb{E}[\hat{\sigma}_n^2(\lambda)] = \sigma_{\text{irr}}^2(\lambda)$

Similarly, for test data, Proposition 3.1 gives $\mathbb{E}[G_m^{\text{test}}(\lambda)] = 0$, so:

$$\mathbb{E}[L^{\text{test}}(\lambda)] = \mathbb{E}[E^{\text{test}}(\lambda)] + \sigma_{\text{irr}}^2(\lambda) \tag{69}$$

Taking the difference:

$$\mathbb{E}[\Delta(\lambda)] = \mathbb{E}[L^{\text{test}}(\lambda) - L^{\text{train}}(\lambda)] = 2\mathbb{E}[G_n^{\text{train}}(\lambda)] \tag{70}$$

A.3.9. MAIN RESULT

*Proof of Theorem 4.2.* From Lemma A.5 and the relation (70):

$$\mathbb{E}[G_{n,j}^{\text{train}}(\lambda)] = \frac{1}{2}\mathbb{E}[\Delta_j(\lambda)] = \sigma_\epsilon^2(\lambda) \cdot \frac{n-1}{n(n-2)} \tag{71}$$

Since the $d$ coordinates are independent:

$$\mathbb{E}[G_n^{\text{train}}(\lambda)] = \sum_{j=1}^{d} \mathbb{E}[G_{n,j}^{\text{train}}(\lambda)] = d \cdot \sigma_\epsilon^2(\lambda) \cdot \frac{n-1}{n(n-2)} = \sigma_{\text{irr}}^2(\lambda) \cdot \frac{n-1}{n(n-2)} \tag{72}$$

$\square$

**A.4. Proof of Corollary 4.3: Peak at minimal linear information)**

*Proof.* Since $\frac{n-1}{n(n-2)} > 0$ for $n > 2$, maximising $\mathbb{E}[G_n^{\text{train}}(\lambda)]$ is equivalent to maximising $\sigma_{\text{irr}}^2(\lambda)$. From Theorem 4.2:

$$\sigma_{\text{irr}}^2(\lambda) = d\left(\sigma_0^2 + \sigma_1^2 - \frac{c(\lambda)^2}{\phi(\lambda)}\right) \tag{73}$$

Since $\phi(\lambda) > 0$, this is maximised when $c(\lambda) = 0$. Solving $c(\lambda) = \lambda\sigma_1^2 - (1-\lambda)\sigma_0^2 = 0$ gives $\lambda^* = \sigma_0^2/(\sigma_0^2 + \sigma_1^2)$.

In the isotropic case, $\|C(\lambda)\|_F^2 = d \cdot c(\lambda)^2$, so $\lambda^*$ coincides with $\lambda_F^*$ from Proposition 4.1. $\square$

## A.5. Proof of Corollary 4.4: Boundary behavior)

*Proof.* At $\lambda = 0$: $c(0)^2/\phi(0) = \sigma_0^4/\sigma_0^2 = \sigma_0^2$. At $\lambda = 1$: $c(1)^2/\phi(1) = \sigma_1^4/\sigma_1^2 = \sigma_1^2$. At $\lambda^*$: $c(\lambda^*) = 0$, so $c(\lambda^*)^2/\phi(\lambda^*) = 0$.

Since $\sigma_{\mathrm{irr}}^2(\lambda) = d(\sigma_0^2 + \sigma_1^2 - c(\lambda)^2/\phi(\lambda))$, it is minimised when $c(\lambda)^2/\phi(\lambda)$ is maximised, which occurs at the boundaries.

$\square$

**Corollary A.6** (Boundary and peak values)**.**

$$\sigma_{\mathrm{irr}}^2(0) = d\sigma_1^2 \tag{74}$$

$$\sigma_{\mathrm{irr}}^2(1) = d\sigma_0^2 \tag{75}$$

$$\sigma_{\mathrm{irr}}^2(\lambda^*) = d(\sigma_0^2 + \sigma_1^2) \tag{76}$$

*When $\sigma_0 = \sigma_1$, we have $\lambda^* = 1/2$ and $\sigma_{\mathrm{irr}}^2(\lambda^*) = 2\sigma_{\mathrm{irr}}^2(0) = 2\sigma_{\mathrm{irr}}^2(1)$.*

*Proof.* At $\lambda = 0$: $c(0) = -\sigma_0^2$, $\phi(0) = \sigma_0^2$, so $c(0)^2/\phi(0) = \sigma_0^2$ and $\sigma_{\mathrm{irr}}^2(0) = d(\sigma_0^2 + \sigma_1^2 - \sigma_0^2) = d\sigma_1^2$.

At $\lambda = 1$: $c(1) = \sigma_1^2$, $\phi(1) = \sigma_1^2$, so $c(1)^2/\phi(1) = \sigma_1^2$ and $\sigma_{\mathrm{irr}}^2(1) = d(\sigma_0^2 + \sigma_1^2 - \sigma_1^2) = d\sigma_0^2$.

At $\lambda^*$: $c(\lambda^*) = 0$, so $\sigma_{\mathrm{irr}}^2(\lambda^*) = d(\sigma_0^2 + \sigma_1^2)$. $\square$

**Corollary A.7** (Asymptotics)**.** *For large $n$:*

$$\mathbb{E}[G_n^{\mathrm{train}}(\lambda)] \approx \frac{\sigma_{\mathrm{irr}}^2(\lambda)}{n} \tag{77}$$

## A.6. Proof of Proposition 4.6: Shared statistics or $r$ and $\epsilon$

*Proof.* For $r$: $(A(\lambda), b(\lambda))$ minimise $\mathbb{E}[\|v^* - Ax - b\|^2]$. The first-order conditions yield $\mathbb{E}[r] = 0$ and $\mathbb{E}[r \cdot X_\lambda^\top] = 0$.

For $\epsilon$: By the definition of conditional expectation, $\mathbb{E}[\epsilon|X_\lambda] = 0$, which implies $\mathbb{E}[\epsilon] = 0$ and $\mathbb{E}[\epsilon \cdot X_\lambda^\top] = 0$. $\square$

# B. Ablations details

## B.1. Datasets

### B.1.1. MAESTRO V3

MAESTRO v3 (MIDI and Audio Edited for Synchronous TRacks and Organisation) (Hawthorne et al., 2019) contains approximately 200 hours of classical piano performances recorded during international piano competitions. The dataset comprises 1,282 compositions divided into train (967 pieces, 154h), validation (137 pieces, 20h), and test (178 pieces, 26h) partitions, representing a 76%/11%/13% split.

**Technical characteristics.** Audio is provided as uncompressed WAV, 16-bit PCM at 44.1 kHz (some tracks at 48 kHz). The total size is approximately 120 GB. The repertoire spans classical music from the baroque to contemporary periods (Bach, Mozart, Beethoven, Chopin, Liszt, Debussy, etc.), with homogeneous professional studio recording quality.

**Preprocessing.** Audio files are resampled to 44.1 kHz mono and segmented into non-overlapping 5 second chunks; partial chunks shorter than 5 seconds are discarded. Each chunk is encoded using Music2Latent (Pasini et al., 2024), yielding latents of dimension $64 \times 50$ (64 channels at 10 Hz temporal resolution). We apply z-score normalisation per channel, with statistics computed on the training set and applied to all splits to prevent data leakage.

**Split methodology.** The split ensures no composition appears in multiple subsets, even when performed by different pianists. This prevents data leakage at the composition level and ensures train and test sets share the same musical distribution, satisfying Assumption 3.3. The homogeneity of the dataset (classical piano only) makes it well-suited for studying memorisation, as the concentrated distribution leaves stronger per-sample imprints.

### B.1.2. MTG-JAMENDO

MTG-Jamendo (Bogdanov et al., 2019) contains over 55,000 tracks representing approximately 3,777 hours of music. The dataset covers approximately 16,000 unique artists and 18,000 albums from more than 150 countries. Tracks come from the Jamendo platform under Creative Commons licences. We use the official *genre-split-0* with a 60%/20%/20% train/validation/test partition.

**Technical characteristics.** Audio is provided as MP3 at 320 kbps, with a variable sample rate (mainly 44.1 kHz). The total size is approximately 500 GB. The dataset spans productions from 2005 to 2020 across all contemporary genres (electronic, rock, pop, jazz, hip-hop, folk, metal, etc.). Unlike MAESTRO, the production quality varies from home-studio to professional recordings. The dataset includes hierarchical multi-label annotations: 87 genres, 40 instruments, and 56 mood/theme tags.

**Preprocessing.** Same pipeline as MAESTRO: resampling to 44.1 kHz mono, segmentation into 5-second chunks, Music2Latent encoding to $64 \times 50$ latents, and z-score normalisation per channel with training set statistics.

**Split methodology.** The split provided uses random sampling with artist stratification only: no artist appears in multiple subsets, ensuring the model is evaluated on artists unseen during training. To satisfy Assumption 3.3, we performed subsampling on the train and test sets with genre stratification, ensuring balanced genre proportions across splits.

### B.1.3. FREE MUSIC ARCHIVE (FMA)

FMA Large (Defferrard et al., 2017b) contains 106,574 clips of 30 seconds each, representing approximately 883 hours of music under a Creative Commons licence. The dataset is organised into subsets of increasing size; we use FMA Large for maximum diversity, which included 161 different genres.

**Technical characteristics.** Audio is provided as MP3 at a constant 320 kbps, with a variable sample rate (mainly 44.1 kHz). The total size is approximately 93 GB. Clips are central excerpts from complete tracks, spanning productions from 2006 to 2017. Quality varies across independent productions but is generally good.

**Preprocessing.** Same pipeline as MAESTRO: resampling to 44.1 kHz mono, segmentation into 5-second chunks, Music2Latent encoding to $64 \times 50$ latents, and z-score normalisation per channel with training set statistics.

**Split methodology.** The official split uses genre stratification with artist separation: (1) genre proportions are maintained across train/validation/test, and (2) no artist appears in multiple sets. This controlled, genre-balanced design contrasts with MTG-Jamendo's natural distribution and satisfies Assumption 3.3.

### B.1.4. CELEBA

CelebA (CelebFaces Attributes) (Liu et al., 2015) contains 202,599 celebrity face images, each annotated with 40 binary facial attributes (e.g., Male, Smiling, Eyeglasses, Young). We use the official split from Hugging Face (`flwrlabs/celeba`): 162,770 train, 19,867 validation, and 19,962 test images.

**Technical characteristics.** Original images are JPEG at approximately 178×218 pixels. The dataset provides binary labels for 40 attributes covering facial features, accessories, and demographics.

**Preprocessing.** Images are resized to 256×256 using bilinear interpolation followed by centre cropping. Pixel values are normalised to $[-1, 1]$. Each image is encoded using the Stable Diffusion VAE (`sd-vae-ft-mse`), yielding latents of dimension $4 \times 32 \times 32$ (4 channels at 32×32 spatial resolution).

**Split methodology.** The official split uses random partitioning of images; the same identity may appear in both the train and test sets. This does not violate Assumption 3.3, which requires the train and test sets to follow the same distribution $p_1$; both are random samples from the same population. However, identity leakage may amplify the membership signal compared to stricter identity-based splits, as the model could encode identity-specific features that are shared across sets.

## B.2. Latents

### B.2.1. MUSIC2LATENT

Music2Latent (Pasini et al., 2024) is a consistency autoencoder for audio compression, designed for efficient generative modelling and Music Information Retrieval (MIR) tasks. Unlike multi-stage approaches or slow iterative sampling methods, Music2Latent achieves high-fidelity single-step reconstruction through end-to-end training with a single consistency loss.

**Architecture.** The model consists of three components: (1) an encoder that downsamples complex-valued STFT spectrograms into a sequence of 64-dimensional latent vectors, using `tanh` activation to constrain representations to $[-1, 1]$; (2) a decoder that upsamples latent vectors with cross connections to the consistency model; and (3) a consistency model based on the NCSN++ UNet architecture that reconstructs the original spectrogram. Key innovations include frequency-wise self-attention to capture long-range frequency dependencies and adaptive frequency scaling to handle varying value distributions across frequencies.

**Compression characteristics.** Audio at 44.1 kHz is compressed to approximately 10 Hz temporal resolution with 64 channels, achieving a $4096\times$ compression ratio. For our 5-second audio chunks, this yields latent representations of dimension $64 \times 50$.

**Relevance to our assumptions.** Although Music2Latent is not a VAE and does not use explicit KL regularisation, the `tanh` activation constrains latent values to a bounded range $[-1, 1]$. Combined with the high compression ratio, this encourages approximately Gaussian marginal distributions in the latent space, as verified empirically in Table 1. The bounded symmetric activation discourages heavy tails and extreme correlations, supporting the approximate isotropy assumed in our theoretical analysis.

### B.2.2. STABLE AUDIO VAE

The Stable Audio VAE (Evans et al., 2024) is the autoencoder component of Stable Audio Open, a text-to-audio generation system developed by Stability AI. Unlike Music2Latent, which uses consistency models, this is a traditional variational autoencoder with explicit KL regularisation toward a Gaussian prior.

**Architecture.** The model uses a fully-convolutional architecture (AutoencoderOobleck) based on the Descript Audio Codec encoder and decoder. The encoder compresses stereo waveforms at 44.1 kHz through five convolutional blocks with strided convolutions for downsampling. The bottleneck is parameterized as a VAE with a latent size of 64 channels. The decoder mirrors the encoder structure using transposed strided convolutions for upsampling. All convolutions are weight normalised.

**Training.** The VAE is trained with three loss terms: (1) a reconstruction loss based on perceptually weighted multi-resolution STFT, handling stereo via mid-side and left-right representations; (2) an adversarial loss with feature matching using 5 convolutional discriminators; and (3) a KL divergence loss regularising the latent distribution toward a standard Gaussian prior. Training was performed on approximately 486,000 audio recordings from Freesound and the Free Music Archive, all under Creative Commons licences.

**Compression characteristics.** Audio at 44.1 kHz is compressed to a latent rate of 21.5 Hz with 64 channels. For our experiments, we convert audio to mono before encoding.

**Relevance to our assumptions.** The explicit KL regularisation toward $\mathcal{N}(0, I)$ directly encourages the latent space to satisfy the Gaussian isotropic assumptions of our theoretical analysis. As shown in Table 1, the Stable Audio VAE latents exhibit low skewness ($\overline{|\gamma|} = 0.08$), low excess kurtosis ($\overline{|\kappa|} = 0.10$), and weak inter-dimension correlations ($\overline{|\rho|} = 0.16$), confirming approximate Gaussianity and isotropy.

### B.2.3. STABLE DIFFUSION VAE

The Stable Diffusion VAE (`sd-vae-ft-mse`) (Blattmann et al., 2022) is the autoencoder component of the Stable Diffusion image generation system. We use the fine-tuned version released by Stability AI, which improves face reconstruction

compared to the original model.

**Architecture.** The model is a KL-regularised autoencoder (kl-f8) with an $8\times$ spatial downsampling factor. The encoder uses convolutional blocks with residual connections to compress images into a latent space with 4 channels. For $256\times256$ input images, this yields latent representations of dimension $4 \times 32 \times 32$. The decoder mirrors the encoder structure using transposed convolutions for upsampling.

**Training.** The original kl-f8 autoencoder was trained on OpenImages with L1 reconstruction loss, LPIPS perceptual loss, and KL divergence regularisation. The `ft-mse` variant was fine-tuned from this checkpoint on a 1:1 ratio of LAION-Aesthetics and LAION-Humans datasets for an additional 280k steps, with increased emphasis on MSE reconstruction (MSE + $0.1 \times$ LPIPS). This fine-tuning improves reconstruction quality, particularly for human faces.

**Compression characteristics.** Images at $256\times256$ pixels are compressed to $4 \times 32 \times 32$ latents, achieving a $48\times$ compression ratio (from $256 \times 256 \times 3 = 196,608$ to $32 \times 32 \times 4 = 4,096$ values).

**Relevance to our assumptions.** Despite the KL regularisation, the Stable Diffusion VAE latent space deviates significantly from the Gaussian isotropic assumptions. As shown in Table 1, CelebA latents encoded with this VAE exhibit high excess kurtosis ($\overline{|\kappa|} = 0.71$), indicating heavy-tailed marginal distributions and strong inter-dimension correlations ($\overline{|\rho|} = 0.61$). These violations explain why the peak prediction $\lambda_F^*$ fails to match the observed peak for this configuration (Table 2), while the bell-shaped curve still appears, confirming that the bell shape is universal but the closed-form peak location requires our assumptions to hold.

## B.3. Model Architectures

We use two backbone architectures, a Transformer and a UNet, to verify that our findings are architecture-independent. Both are trained with the Rectified Flow objective (Liu et al., 2023; Lipman et al., 2023).

### B.3.1. TRANSFORMER (DIT)

Our Transformer follows the Diffusion Transformer (DiT) architecture (Peebles & Xie, 2023) with several modifications for audio sequences.

**Architecture.** The input sequence $(B, C, T)$ is first transposed to $(B, T, C)$ and projected to the hidden dimension via a linear layer. Each Transformer block consists of:

- **Attention**: Multi-head self-attention with Rotary Position Embeddings (RoPE) (Su et al., 2024) and Flash Attention (Dao et al., 2022) for efficiency.
- **MLP**: Two-layer feedforward network with GELU activation (tanh approximation).
- **adaLN-Zero conditioning**: Adaptive Layer Normalisation with six modulation parameters (scale, shift, and gate for both attention and MLP branches), initialised to zero for stable training.

Time conditioning uses sinusoidal embeddings processed through a two-layer MLP with SiLU activation. The final layer applies adaLN modulation followed by a linear projection back to the input dimension.

**Configuration.** For audio experiments, we use the 410M parameters configuration: hidden size 576, depth 24, 12 attention heads, and an MLP ratio of 4.0. Initialisation follows Xavier uniform for linear layers, with zero initialisation for all adaLN modulation layers and the final output projection.

### B.3.2. UNET

We implement UNet architectures for both 1D audio latents and 2D image latents, sharing the same structural design.

**Architecture.** The UNet follows a symmetric encoder-decoder structure with skip connections:

- **Encoder**: Sequence of ResBlocks at each resolution level, with strided convolutions for downsampling between levels.

- **Middle**: ResBlock → Self-Attention → ResBlock at the lowest resolution.

- **Decoder**: Sequence of ResBlocks with skip connections from the encoder, with nearest-neighbour upsampling followed by convolution between levels.

Each ResBlock consists of: GroupNorm → SiLU → Conv → time conditioning → GroupNorm → SiLU → Dropout → Conv, with a residual connection. Time conditioning injects the timestep via scale and shift modulation: $h \leftarrow h \cdot (1 + \text{scale}) + \text{shift}$, where scale and shift are produced by an MLP from the sinusoidal time embedding. Self-attention blocks use GroupNorm followed by multi-head attention. All output convolutions and attention projections are zero-initialised for stable training.

**Configuration.** For the medium configuration:

- **UNet 1D** (audio, $64 \times T$ latents): base channels 192, channel multipliers $(1, 2, 4, 4)$, 2 ResBlocks per level, attention at levels 2–3, dropout 0.1.

- **UNet 2D** (images, $4 \times 32 \times 32$ latents): base channels 192, channel multipliers $(1, 2, 4, 4)$, 2 ResBlocks per level, attention at levels 1–3, no dropout.

### B.3.3. TRAINING

All models are trained with the AdamW optimiser ($\beta_1 = 0.9$, $\beta_2 = 0.999$), a learning rate of $10^{-4}$, mixed-precision (FP16/BF16), and gradient clipping set at 1.0. We use early stopping based on validation loss with a patience of 25 epochs. The batch size is 128 for Transformer models and 64 for UNet models.

**Relevance to our analysis.** As shown in Table 2, both architectures yield nearly identical observed peak locations $\lambda_{\text{obs}}$ on the same dataset, confirming that $\lambda_F^*$ depends on data geometry $(\Sigma_0, \Sigma_1)$ rather than model architecture or capacity.

## C. Additional Metrics Analysis

In the main text, we report the train-test gap using the mean reconstruction error. Our protocol evaluates each sample with $K = 100$ independent noise realisations, yielding a distribution of reconstruction errors per sample rather than a single value. This enables the computation of richer statistics, including the median, quartiles, and standard deviation, which provide robustness cheques.

Here we examine these additional metrics to assess the robustness of our findings. Results are presented on MTG-Jamendo; similar patterns hold for MAESTRO v3 and FMA Large.

### C.1. Median and Quantile Metrics

Figure 8 shows the normalised gap for the median and quartiles ($q^{0.25}$, $q^{0.75}$). All three metrics exhibit bell-shaped curves nearly identical to the mean, with peaks at $\lambda = 0.5$ and boundary values approaching zero.

This consistency across robust statistics confirms that the bell-shaped pattern is not driven by outliers. The membership signal is present throughout the distribution of reconstruction errors, not just in the tails. This robustness further supports our theoretical framework: the $\lambda$-dependent structure of the train-test gap is a fundamental property of the model, not a statistical artefact.

### C.2. Standard Deviation: An S-Shaped Pattern

The standard deviation across the $K = 100$ noise samples reveals a qualitatively different pattern. As shown in Figure 8, instead of a bell curve, we observe an S-shaped curve:

- For $\lambda < 0.3$: the gap is negative, meaning training samples exhibit *higher* variance in reconstruction error than test samples.

- For $\lambda > 0.3$: the gap becomes positive, meaning training samples exhibit *lower* variance.

**Interpretation.**    We hypothesise that this pattern reflects sample-specific attractors in the learnt velocity field. At low $\lambda$ (high noise), training samples may either be captured by their learnt attractor or missed entirely, producing high variance across noise realisations. Test samples, lacking specific attractors, consistently receive population-average predictions with lower variance. As $\lambda$ increases and samples approach the data manifold, training samples reliably reach their attractors (low variance), while test samples show more variable behaviour.

This interpretation remains speculative; a theoretical characterisation of higher-order statistics is left for future work.

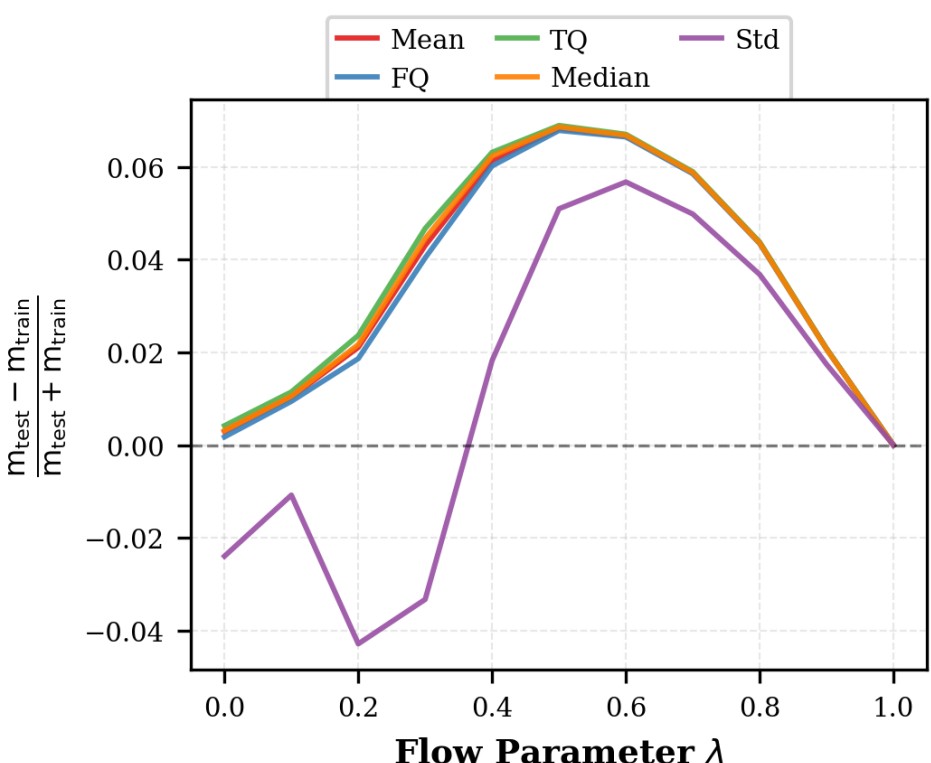

*Figure 8.* Normalised gap for all metrics on MTG-Jamendo. Mean, median, and quartiles ($q^{0.25}$, $q^{0.75}$) exhibit consistent bell-shaped curves. Standard deviation ($\sigma$) shows an S-shaped pattern. Similar patterns are observed on MAESTRO v3 and FMA Large.

## D. Failure Modes and Relaxation of Assumptions

### D.1. Limits of Controlled Perturbations

During the rebuttal period, we explored controlled transformations to test assumption boundaries: $z \mapsto \text{sign}(z)|z|^p$ to modulate kurtosis and $z \mapsto (1 - \alpha)z + \alpha \cdot \text{mean}(z)$ to inject inter-dimension correlations. However, these transformations introduce auxiliary artefacts and perturb multiple statistics simultaneously, making clean isolation impossible; therefore, we do not rely on them. We rely instead on naturally distinct configurations (different datasets and encoders), each producing their own $(\Sigma_0, \Sigma_1)$ pairs and degrees of assumption violation (Table 1). The CelebA / SD VAE configuration, with $\overline{|\rho|} = 0.61$ and $\overline{|\kappa|} = 0.71$, serves as our primary natural test case.

Based on our theoretical analysis, we interpret the failure modes as follows. Non-Gaussianity introduces a nonlinear residual $r(x, \lambda)$ (Section 4.3) that shifts the irreducible variance non-uniformly across $\lambda$, displacing the peak from $\lambda_F^*$. Anisotropy causes $\text{tr}(\Sigma_1^2)$ in the denominator of Proposition 4.1 to be dominated by off-diagonal entries, pushing $\lambda_F^*$ downward, consistent with the CelebA case where $\lambda_F^* = 0.45$ while $\lambda_{\text{obs}} \in [0.6, 0.7]$.

In practice, we recommend that practitioners directly measure the bell-shaped curve on their dataset of interest: computing $\lambda_F^*$ requires only $O(d^2)$ trace estimations, and observing the empirical peak requires only forward passes at a grid of $\lambda$ values.

Checking whether the two agree is both inexpensive and more informative than any synthetic perturbation experiment.

## D.2. Relaxation: From $\lambda_F^*$ to $\lambda_{\text{irr}}$

A natural alternative to the closed-form $\lambda_F^*$ is to numerically minimise

$$\sigma_{\text{irr}}^2(\lambda) = \text{tr}(\Sigma_V) - \text{tr}(C(\lambda)\Phi(\lambda)^{-1}C(\lambda)^{\top}) \tag{78}$$

over $\lambda$, yielding a $\lambda_{\text{irr}}$ that does not require the isotropy assumption.

However, for the high-dimensional latents commonly considered ($d = 64 \times 50 = 3200$ for audio), reliably estimating and inverting $\Phi(\lambda) \in \mathbb{R}^{d \times d}$ from finite samples is, in practice, unstable, as audio chunks are temporally correlated and the effective sample size is much smaller than the nominal $n$.

The closed-form $\lambda_F^*$, in contrast, depends only on the traces of products of $\Sigma_0$ and $\Sigma_1$, which can be estimated robustly in $O(d^2)$ time.

## E. Membership Inference Attack Details

**Feature extraction.** For each sample $x_1$, we compute reconstruction errors at each $\lambda \in \{0, 0.1, \ldots, 1.0\}$ using $K = 100$ independent noise realisations, and extract the per-$\lambda$ mean, yielding an 11-dimensional feature vector.

**Classifier and training.** We train a small MLP (2 hidden layers, 64-32 units) on the $\lambda$-resolved features using binary cross-entropy and Adam ($\beta_1 = 0.9$, $\beta_2 = 0.999$), with early stopping on a held-out validation set. The architecture was selected via Bayesian optimisation over depth, width, and training duration.

**Dataset construction.** We partition the generative model's training and held-out sets into two disjoint halves each, combining one half of each for MLP training and the other for evaluation, ensuring no sample appears in both. We use 1,000 samples per class for training and 500 for testing.

**Results.** Table 3 reports AUC and TPR@5%FPR across all datasets and baselines. Figure 9 shows the confusion matrix at threshold 0.38.

Table 3. MIA results across datasets. AUC with TPR@5%FPR (%) in parentheses.

| METHOD | MAESTRO V3 | MTG-JAMENDO | FMA LARGE | CELEBA |
|---|---|---|---|---|
| NAIVE$_{\text{RF}}$ | 0.67 (14.1) | 0.57 (6.0) | 0.55 (4.8) | 0.58 (8.0) |
| SECMI$_{\text{RF}}$ | 0.72 (13.9) | 0.61 (11.0) | 0.59 (8.4) | 0.56 (4.3) |
| PIA$_{\text{RF}}$ | 0.83 (36.5) | 0.64 (10.2) | 0.61 (9.3) | 0.62 (14.0) |
| OURS | **0.91** (56.7) | **0.72** (23.4) | **0.67** (19.0) | **0.65** (15.0) |

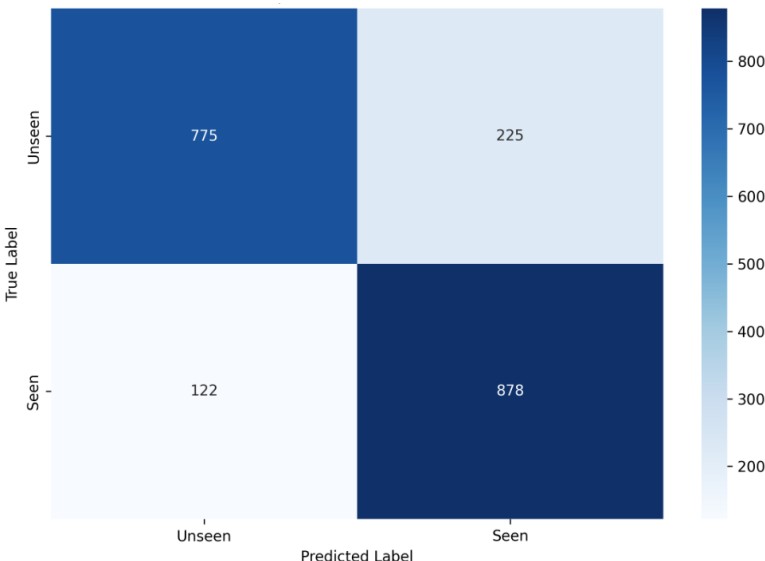

*Figure 9.* Confusion matrix on MAESTRO at threshold 0.38. The classifier correctly identifies 82% of members and 84% of non-members.

## F. Reflow: Preliminary Results

The reflow procedure (Liu et al., 2023) replaces the independent coupling $X_0 \perp\!\!\!\perp X_1$ with learnt pairs obtained by integrating the trained velocity field forward from noise samples. This breaks the independence assumption underlying our theoretical analysis, and we conjecture it attenuates the membership signal by correlating the noise endpoint with the data.

**Protocol.** We train a single reflow step on MAESTRO v3, using the same Transformer architecture (410M parameters) and training hyperparameters as the baseline configuration (Section 5). The reflow pairs $(x_0, x_1)$ are obtained by integrating the baseline model forward from $x_0 \sim \mathcal{N}(0, \Sigma_0)$.

**Results.** Figure 10 shows the normalised train-test gap $\Delta_{\mathrm{norm}}(\lambda)$ for the reflow model alongside the baseline. The bell-shaped structure persists, confirming that the phenomenon is not specific to the independent coupling. However, the peak magnitude decreases substantially (from 0.09 to 0.01), and the curve exhibits a broader, flatter plateau rather than a sharp peak. The peak location remains near $\lambda_F^*$, consistent with the interpretation that the peak is governed by data geometry rather than the coupling procedure.

These results also suggest that reflow may offer a natural mitigation of membership leakage as a byproduct of its trajectory-straightening objective; though a thorough characterisation is left for future work.

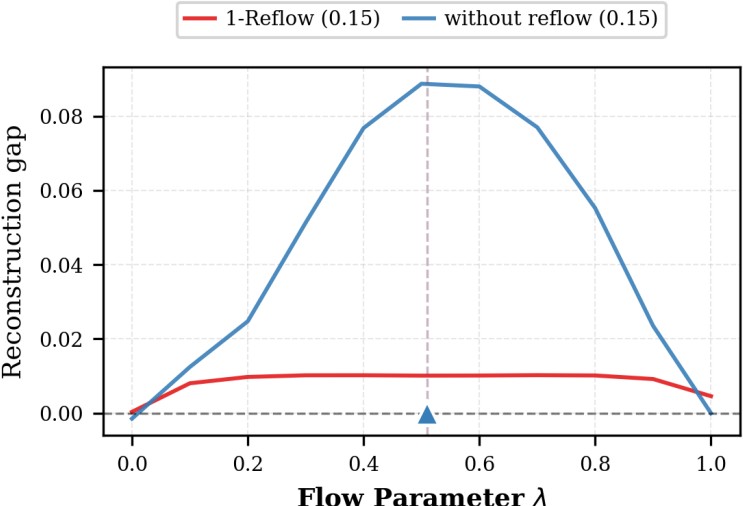

*Figure 10.* Normalised train-test gap $\Delta_{\mathrm{norm}}(\lambda)$ for the baseline and reflow models on MAESTRO v3. The bell shape persists under reflow but with a substantially reduced magnitude and broader plateau.

