# OpenReview forum: "Where Rectified Flows Leak: Characterising Membership Signals Along the Interpolation Path"
_ICML.cc/2026/Conference — ICML 2026 regular_

### Official Review · Reviewer_gSk2 · 2026-02-28

**Soundness:** 3
**Presentation:** 4
**Significance:** 3
**Originality:** 4
**Overall Recommendation:** 5
**Confidence:** 4

**Summary:**

The authors investigate the vulnerability of Rectified Flow models to MIAs. They state that models leak membership signals unevenly along the interpolation path. By decomposing the training and test losses, they theoretically demonstrate that the train-test gap follows a bell-shaped curve over the trajectory. Under strong assumptions, they derive a closed-form solution for the peak of this curve. They empirically validate this phenomenon on audio and image datasets. Finally, they use the reconstruction errors as features for a white-box MIA, achieving an AUC of 0.91.

**Compliance With Llm Reviewing Policy:**

Affirmed.

**Final Justification:**

The initial manuscript was slightly below the acceptance threshold but the rebuttal clarified my concerns. The rebuttal discussion recommendations should be incorporated in the manuscript. In my opinion, the work is sufficiently interesting, technically sound and novel to meet the criteria for acceptance.

**Key Questions For Authors:**

- Please provide baselines for the MIA attack in terms of SOTA MIA like SecMI (Duan et al.) and against CelebA
- Theorem 4.2 solves an OLS problem. Can you provide evidence showing how much the residual actually contributes to the loss at the peak in the trained models?
- Impact of Reflow: Can you provide a small-scale empirical test showing what happens to the train/test gap if you use even a single step of Reflow? Does the bell shape collapse completely?

**Limitations:**

yes

**Strengths And Weaknesses:**

Strengths:
- Probing the interpolation trajectory for localised “memorisation signals” is an intuitive and original approach to privacy analysis that provides a more granular understanding than aggregate privacy metrics
- The mathematical decomposition into irreducible variance, approximation error and membership signal is well-constructed.
- The authors perform extensive ablations with varous data distributions, noise distributions, encoders and architectures
- The authors are honest about limitations where violations of their isotropic Gaussian assumption leads to predictions failing.

Weaknesses:
- The closed-form solution in Theorem 4.2 relies on two very strong assumptions, that the latent space is isotropic Gaussian and that the optimal predictor is a linear OLS estimator. Sections 4.3 and 4.4 attempt to wave the linearity assumption away but does not provide a formal argument or proof for linearity. There is a huge difference between a linear regression proof and a Diffusion Transformer
- The authors actually admit that the theoretical prediction fails on CelebA at the beginning of Section 6.1 because of anisotropy. This suggests that the closed-form is fragile and the theory fails exactly in one of the most important applications of generative models: high-resolution images
- The MIA attack has no baselines at all. It’s impossible to properly evaluate whether the $\lambda$-specific MIA provides any benefits without understanding how it would compare e.g. to naive thresholding on the reconstruction error or MIA techniques adapted for Diffusion Models.
- Moreover, MIA is performed only on MAESTRO, which fits the theoretical assumptions best. The important case would be to understand what happens when the assumptions fail, such as CelebA. Does the attack still achieve high AUC (or better still: High TPR at low FPR) when the peak is no longer where the theory predicts it to be?
- Applications to larger models are required, the small custom-trained models are unable to show that the technique is actually applicable to real-world applications
- The authors explicitly exclude Reflow. However, Reflow is probably the most important technique for fast sampling. Excluding it severely limits the applicability of this work

---

> ### Author Rebuttal · Authors · 2026-03-30
>
> We sincerely thank the reviewer for their detailed and constructive feedback which significantly strengthened our work. We conducted new experiments during the rebuttal period addressing each concern, baselines, broader evaluation, the OLS-to-Transformer gap, and reflow.  We summarize our findings below.
>
> ---
> ## Key Questions
>
> **Q1.** _Baselines for the MIA attack (SecMI, PIA) and results on CelebA?_
>
> Following reviewer suggestion, we implemented baselines.
> Our method outperforms baselines on every dataset.
> SecMI reduces to single-λ velocity error in RFs (forward path analytically known), equivalent to the Naive attack. PIA was adapted using model-estimated trajectories.
> Notably, the Naive consistently peak at $λ^*_F$.
>
> |Dataset (sample size)|Ours (λ-resolved structure)|PIA_RF|Naive/SecMI_RF|
> |-|-|-|-|
> | MAESTRO(1K)|**0.91** (0.57)|0.83 (0.17)|0.67 (0.14)|
> | MTG-Jamendo(55K) |**0.72** (0.23)|0.64 (0.10)|0.58 (0.06)|
> | FMA Large(106K)|**0.67** (0.19)|0.61(0.09)|0.55 (0.08)|
> | CelebA(162K)|**0.65** (0.15)|0.62 (0.14)|0.58 (0.08)|
> *Table 1: AUC Score on MIA task and TPR@5% in parenthesis*
>
> Results will be included in section 7) on MIA application.
>
> *Remark:* Our MIA uses the full λ-resolved profile, not closed-form $λ^*_F$; the attack remains effective whenever the bell shape holds, including CelebA (0.65 AUC).
> Beyond MIA itself, we believe our primary contribution is the theoretical characterization of where and why memorizaton signals arise. It gives insights into how learning operates in RF, opening potential applications for guided defenses, privacy-preserving training, or tailored training.
>
> **Q2.** _How much does the nonlinear residual contribute to the loss at the peak?_
>
> We provide empirical evidence that linear information vanishing at  $λ^*_F$​ governs the Transformer's memorization in practice.
>
> We trained an OLS model on the same RF task and compared it to the full Transformer.
> The Transformer/OLS test loss ratio equals ≈1.0 at $λ ∈ {0,1}$ (both models perform identically at the boundaries) and peaks at 2.14 near λ^\*\_F (on MAESTRO/M2L with Σ₀ × 1). The pattern shifts consistently when Σ₀ is scaled (×0.25, ×4), matching λ^\*\_F consistently. We observe the same behavior on MTG-Jamendo.
> This validates the core mechanism: at the boundaries, linear information dominates; near $λ^*_F$, the Transformer must rely on nonlinear, sample-specific features, exactly where the membership signal peaks.
>
> These analyses will be integrated into an extended Section 4.4.
>
> **Q3.** _Impact of Reflow: does the bell shape collapse?_
>
> No, the bell shape persists. We trained a model with one step of reflow on MAESTRO. The bell-shaped curve remains, with a broader, flatter plateau rather than a sharp peak, and the maximum gap magnitude decreases (from 0.07 to 0.015), yet the center of the plateau remains on $λ^*_F$.
> These results will be added to the ablations section.
>
>
> ## Weaknesses
>
> **W2.** _Theory fails on CelebA_
>
> Following reviewer comment on this weaknesses, we have now characterized precisely why it fails, separating two effects. We conducted controlled perturbation experiments on MAESTRO/ (fast iteration), independently varying each assumption.
>
> **Gaussianity (kurtosis):** We applied $z→sign(z)|z|^p$ to vary kurtosis (up to 20× baseline) while keeping correlations stable. The peak shifts by at most −0.15 proportionally to the kurtosis factor; the bell shape is fully preserved. The closed-form $λ^*_F$ breaks for $|\kappa| > 0.4$. Theoretically, non-Gaussianity introduces a residual r(x,λ) reducing effective irreducible variance non-uniformly in λ, shifting the peak.
>
> **Correlations:** We injected inter-channel correlation via $z→(1−α)z+α·mean_{channels}(z)$. The closed-form $λ^*_F$ breaks for $|ρ|>0.6$. Theoretically, anisotropy causes $tr(Σ₁²)$ in the denominator to be dominated by off-diagonal entries, shifting the prediction toward 0.
>
> **Correlation relaxation:** On pratical scenario, we can relax the isotropy assumption by computing Φ⁻¹ numerically (Proposition 4.1) instead of the closed form. We tested this on our correlated perturbation experiments: the numerically predicted $λ_{irr}$ recovers the correct peak at each correlation level, and the analytical $λ^*_F$ matches it when correlation is low.
>
> These analyses will be generalized and integrated into an extended Section 4.4.
>
> **W5.** _Larger models_
>
> We agree our claims were overstated. We trained Transformers on various datasets from 8M to 770M parameters and a UNet, all yielding the same peak for a fixed dataset/encoder. The phenomenon is governed by data geometry, not model capacity, indirect evidence of scalability. Rebuttal time constraints prevented testing on FLUX or SD3, but this is planned as a next step. Following the reviewer's advice, we will reframe the contribution as foundational:characterizing where and why membership signals arise in rectified flows, rather than claiming direct applicability to deployed systems.

---

> > ### Author Rebuttal · Reviewer_gSk2 · 2026-04-01
> >
> > Thank you for your detailed rebuttal. It's obvious that a lot of hard work went into this, and I appreciate the intellectual honesty. Please incorporate the discussed points into the manuscript as proposed.

---

### Official Review · Reviewer_gfyf · 2026-03-12

**Soundness:** 3
**Presentation:** 4
**Significance:** 3
**Originality:** 3
**Overall Recommendation:** 4
**Confidence:** 3

**Summary:**

The authors present a theoretical framework for analyzing membership signals in “Rectified Flows” and demonstrate their findings using Membership Inference Attack experiments over an audio dataset.

**Compliance With Llm Reviewing Policy:**

Affirmed.

**Final Justification:**

The response and discussion with both myself and other reviews re-iterates the value of this paper; I maintain my support for it.

**Key Questions For Authors:**

The key question that would impact scoring would be for the authors to more strongly justify the connection between the theory and experimentation, mainly by better clarifying why the assumptions (most notably the Gaussian latents) made in the theory hold in practice, or showing that the theory might hold with some relaxations.

**Limitations:**

Yes

**Strengths And Weaknesses:**

Strengths
- Thorough ablation study
- Strong convincing demonstrative result of 0.91 AUC
- Well written theoretical section
Weaknesses
- The arguments for why the theoretical assumptions hold in practice seem weak and hand-wavy
- White-box access is a strong assumption to make

---

> ### Author Rebuttal · Authors · 2026-03-30
>
> We sincerely thank the reviewer for their positive assessment and feedback. The key question asks us to justify the connection between theory and experiments, specifically, how the assumptions hold in practice, or whether the theory survives without them. We conducted new experiments during the rebuttal to answer this directly, and provide three layers of evidence below, hoping this addresses the reviewer's remarks.
>
> ## Justifying how the assumptions hold in practice
>
> **Controlled perturbation experiments.** Starting from (MAESTRO/M2L) latents (which satisfy minimal Gaussianity), we independently vary each assumption to map precise failure boundaries.
>
> _Gaussianity (kurtosis):_
> We applied $z → sign(z)|z|^p$ to vary kurtosis (up to 20× initial value) while keeping correlations stable.
>  The peak shifts by at most −0.15 proportionally to the kurtosis factor; the bell shape is fully preserved. The closed-form $λ^*_F$ breaks down above $|\kappa| > 0.4$.
>  Theoretically, non-Gaussianity introduces a nonlinear residual $r(x, λ)$ that reduces the effective irreducible variance non-uniformly in λ, shifting the peak downward, but the effect remains small.
>
> _Correlations:_
> We injected inter-channel correlation via $z →(1−α)z + α·mean_{channels}(z)$, varying $|ρ|$ while keeping kurtosis fixed. The closed-form λ^\*\_F breaks down above $|ρ| > 0.6$; the bell shape remain fully preserved.
> Theoretically, anisotropy causes $tr(Σ₁²)$ in the denominator of $λ^*_F$ formula to be dominated by off-diagonal entries, shifting the prediction toward 0.
>
> **OLS validation.**
> We confirmed memorization peaks where linear information is minimal.
> We trained a single-layer linear predictor (OLS) on the RF task and compared it to the full  Transformer as a function of λ.
> The Transformer/OLS test loss ratio equals ≈1.0 at $λ ∈ {0,1}$ and peaks at λ^\*\_F : near the endpoints, a linear model suffices; at $λ^*_F$, linear information vanishes and the model must rely on nonlinear, sample-specific features. This directly validates the core mechanism of our theory. The observation holds across datasets (MAESTRO, MTG-Jamendo) and shifts consistently when $Σ₀$ is scaled (×0.25, ×4), matching theoretical predictions in each case.
>
> These analyses will be generalized and integrated into an extended Section 4.4.
>
> ## Showing the theory holds with relaxations
>
> **Correlation relaxation:**
> The isotropy assumption can be relaxed numerically: rather than relying on the closed-form λ^\*\_F, one can directly minimize $σ²_{irr}(λ)=tr(Σ_V)−tr(C(λ)Φ(λ)⁻¹C(λ)^⊤)$ to predict $λ_{irr}$ (Proposition 4.1).
> Under isotropy $C(λ_F)=0$ makes the closed-form tight, while in the anisotropic case computing $Φ⁻¹$ numerically bypasses this limitation.
> We validate this on our correlated perturbation experiments: the numerically predicted $λ_{irr}$ recovers the correct peak at each correlation level, with $λ^*_F$ matching it when correlation is low.
>
> This analysis will be generalized and integrated into an extended Section 4.4.
>
>
> ## What is universal vs. what requires the assumptions
>
> **Universal:** The bell-shaped structure, boundary behavior (Corollary 4.4), and temporal accumulation of the signal (Figure 3) hold across all configurations, including CelebA where both assumptions are most violated and previous controled perturbations experiments.
>
> **Relies on assumptions:** Only the closed-form peak location $λ^*_F$ requires these assumptions; and as shown above, the isotropy requirement can be relaxed numerically.
>
>
> *Remark:* Our analyses above is limited to Maestrov3 due to time / ressources constraint during rebutal period. Yet, we planned to generalize our findings to all datasets.
>
> We will add a summary at the end of the experimental section, clarifying what holds and under which conditions.
>
> ## On the white-box assumption
>
> White-box access is indeed a strong assumption. Our goal is not to propose a deployment-ready attack, but to characterize a structural property of rectified flows, where and why membership signals emerge. The MIA serves as a proof of concept that this signal is exploitable. Beyond the MIA itself, we believe our primary contribution is the theoretical characterization of where and why membership signals arise, the bell-shaped structure and $\lambda^*_F$ prediction. It give insights into how learning operates along the rectified flow interpolation path, opening potential applications for guided defenses, privacy-preserving training, or tailored training. We will make this positioning clearer in the revision.

---

> > ### Author Rebuttal · Reviewer_gfyf · 2026-04-03
> >
> > Response addresses concerns; enthusiasm remains at the level it was at the time of reviewing (positive).

---

### Official Review · Reviewer_QZ2i · 2026-03-13

**Soundness:** 2
**Presentation:** 2
**Significance:** 2
**Originality:** 2
**Overall Recommendation:** 3
**Confidence:** 3

**Summary:**

The paper analyzes membership inference in Rectified Flows by studying reconstruction error along the interpolation path $X_\lambda = (1 - \lambda)X_0 + \lambda X_1$. The core finding: the train-test gap in reconstruction error follows a bell-shaped curve over $\lambda$. The authors derive a closed-form expression for the peak location $\lambda^*_F$ under Gaussian isotropic assumptions, validate it across audio and image datasets, and use it to hit 0.91 AUC in a membership inference attack on a piano music dataset.

**Compliance With Llm Reviewing Policy:**

Affirmed.

**Key Questions For Authors:**

1. What is the MIA AUC on FMA Large and MTG-Jamendo? If significantly lower, how should we interpret the practical relevance of the attack?
2. Have you tried a baseline MIA using reconstruction error at a single fixed $\lambda$, e.g. $\lambda = 0.5$? How much does the full 11-dimensional feature vector improve over it?
3. The Discussion suggests the peak can be found on a cheap proxy model and transferred. Have you verified this empirically?
4. How does the signal scale with dataset size? Corollary A.7 predicts $G^{train} \propto 1/n$, suggesting the attack should weaken on larger datasets. Is this consistent with the weaker signal on FMA Large (106K clips vs. MAESTRO's ~1K compositions)?

**Limitations:**

## Limitations

**Gap between heuristic and formal guarantee in the general case.** The transition from the isotropic Gaussian setting to the general case in Section 4.3 relies on arguments about spectral bias and gradient descent dynamics rather than formal derivation. While Proposition 4.5 is correct, the inference that the membership signal therefore peaks at $\lambda^*_F$ requires assumptions about how neural networks allocate representational capacity that are not proven. The paper appropriately labels this reasoning as heuristic, but it does not characterize the conditions under which the heuristic should be expected to fail beyond the vague qualifier ''when Gaussianity breaks.'' A more actionable account of failure modes tied to, e.g., data dimensionality, encoder geometry, or training regime would substantially strengthen the theoretical contribution.

**Absence of MIA baseline comparisons.** Despite citing SecMI, PIA, and Matsumoto et al. in the related work, the paper evaluates MAESTRO-MIA against no existing membership inference attacks. Even a comparison against a single-$\lambda$ threshold baseline would clarify whether the $\lambda$-resolved structure is doing meaningful work or whether any reconstruction-based MIA would achieve similar performance on this target. The reported AUC of 0.91 cannot be interpreted in isolation: without baselines, it is unclear whether this reflects the power of the proposed method or the vulnerability of the target model.

**Evaluation confined to a single, favorable dataset.** MIA performance is reported exclusively on MAESTRO, a small (${\sim}200$ hours), typologically homogeneous (classical standing piano only) dataset encoded with a specific autoencoder. The paper demonstrates that the bell-shaped gap exists across other datasets but does not report AUC or attack performance on any of them. Critically, MAESTRO exhibits the strongest gap magnitude among all datasets evaluated (See Fig. 4), making the 0.91 AUC to be the likely ceiling rather than a representative result. The paper does not report what attack performance looks like on FMA Large, where the gap is visibly weaker, precisely the case where the method's practical limits would become apparent.

**Insufficient coverage of large-scale deployed systems.** The CelebA experiment already surfaces a notable failure case, yet the paper does not extend evaluation to modern large-scale generative systems such as FLUX or Stable Diffusion 3, despite invoking them in the introduction as primary motivation. The gap between "the membership signal is detectable in Music2Latent latents of piano recordings" and "this framework has implications for deployed text-to-image systems" is substantial and remains unaddressed. Until the approach is validated or its limitations clearly characterized on the class of models that motivate the work, the broader relevance of the results is difficult to assess.

**Strengths And Weaknesses:**

## Strengths

**Clean theory.** The decomposition of training loss into approximation error, irreducible variance, and the cross-correlation term $G^{train}_n(\lambda)$ is elegant. Proposition 3.1 shows $G^{train}_n(\lambda)$ vanishes in expectation for test data while being generically non-zero for training data. The closed-form peak $\lambda^*_F = \sigma^2_0 / (\sigma^2_0 + \sigma^2_1)$ in the isotropic case is satisfying: the signal is strongest exactly where linear prediction from $X_\lambda$ to $V$ breaks down.

**A good explanation for why standard metrics miss the signal.** The dual masking mechanism (Section 4.5) is one of the paper's most valuable contributions. Spatial averaging over $\lambda$ dilutes the localized signal, while temporal compensation makes the growing membership signal indistinguishable from improving approximation in the training loss. The temporal evolution plots in Figure 3, showing validation loss decreasing while the gap grows, are particularly convincing.

**Ablations.** Table 2 varies data distribution, noise variance, latent space, architecture, and modality. The key result -- that peak location is architecture-independent but data-geometry-dependent -- is well-demonstrated across all conditions.



## Weaknesses

W1. Section 4.3 transitions from the tight Gaussian isotropic case to the general case via arguments about spectral bias and gradient descent dynamics. Proposition 4.5 is correct. But the leap from that to "therefore the membership signal peaks at $\lambda^*_F$" requires assumptions about how neural networks allocate capacity that aren't proven. The paper is upfront about calling this heuristic, but it needs a more careful account of when the heuristic should be expected to fail, beyond "when Gaussianity breaks."

W2. The paper cites SecMI, PIA, and Matsumoto et al. in related work but never compares against any of them. Even a comparison against a single-$\lambda$ threshold attack, would clarify whether the $\lambda$-resolved structure is doing the work or whether MAESTRO is just an easy target for any reconstruction-based MIA. The 0.91 AUC sounds strong. Without baselines, it's hard to know what it means.

W3. The attack runs on one dataset: MAESTRO, which is small (~200 hours), homogeneous (classical piano only), and encoded with a specific autoencoder. The paper shows the bell-shaped gap exists for other datasets but doesn't report MIA performance on them. MAESTRO also shows the strongest gap magnitude among all datasets (Figure 4), so the 0.91 AUC is likely a ceiling, not a representative result. What AUC does the attack achieve on FMA Large, where the gap is much weaker?

W4. The CelebA experiment already shows a significant failure case. More importantly, the paper doesn't touch modern large-scale systems like FLUX or Stable Diffusion 3, despite mentioning them in the introduction as motivation. The gap between "the theory works on Music2Latent latents of piano recordings" and "this has implications for deployed systems like FLUX" is large and unadressed.

---

> ### Author Rebuttal · Authors · 2026-03-30
>
> We thank the reviewer for these insightful comments, which significantly strengthened our work. We conducted new experiments addressing each concern (adding baselines, broadening the evaluation, and clarifying the heuristic's scope) and summarize at best our findings below, hoping this addresses the reviewer's remarks.
> ## Key Questions
> **Q1.a** _MIA AUC on FMA Large and MTG-Jamendo?_
>
> |Dataset (sample size)|Ours|PIA_RF|Naive/SecMI_RF|
> |-|-|-|-|
> | MAESTRO(1K)|**0.91** (.57)|0.83 (.17)|.67 (.14)|
> | MTG-Jamendo(55K) |**0.72** (.23)|0.64 (.10)|0.58 (.06)|
> | FMA Large(106K)|**0.67** (.19)|0.61(.09)|0.55 (.08)|
> | CelebA(162K)|**0.65** (.15)|0.62 (.14)|0.58 (.08)|
> *Table 1: AUC Score on MIA task and TPR@5% in parenthesis*
>
> *Remark:* Importantly, our MIA uses the full λ-resolved profile, and do not depend on the closed-form $λ^*_F$; the attack remains effective whenever the bell shape holds, including CelebA (0.65 AUC).
>
> **Q1.b** _Practical relevance if significantly lower?_
>
> Our method outperforms all baselines in every configuration. Beyond MIA itself, we believe our primary contribution is the theoretical characterization of where and why memorizaton signals arise. It give insights into how learning operates in RF, opening potential applications for guided defenses, privacy-preserving training, or tailored training.
>
> **Q2.** _Improvement of the full feature vector over a single-λ baseline?_
>
> Up to +24 AUC score on MAESTRO, and consistent improvement across all datasets (Table 1). Following reviewer's suggestion, we adapted PIA and SecMI to RF. SecMI reduces to a single-λ velocity error (equivalent to the Naive attack), and PIA was adapted using model-estimated trajectories. Notably, the Naive consistently peak at $λ^*_F$
>
> Results from Q1 and Q2 will be included in the extended section on MIA applications.
>
> **Q3.** _Empirical verification of proxy model transfer claim?_
>
> Yes. We trained Transformers from 8M to 770M parameters (MAESTRO, MTG-Jamendo, FMA Large) and a UNet (MAESTRO): all yield the same peak location for a fixed dataset/encoder, suggesting the peak is structural to data geometry, and a cheap proxy model can locate λ^\*\_F for larger targets. It will be added as empirical justification in the discussion.
>
> **Q4.** _Signal scaling consistency with $1/n$?_
>
> Yes. The AUC decrease across MAESTRO (1K) $\to$ MTG-Jamendo (55K) $\to$ FMA Large (106K) matches Corollary A.7's prediction $G_n \propto 1/n$. It will be stated explicitly in Section 6.1.
>
> ## Weaknesses
> **W1.** _Heuristic-to-formal gap in Section 4.3_
>
> We conducted controlled perturbation experiments to address this gap. The Gaussian and isotropy assumptions are necessary for the analytical peak location formula, but not for the bell shape. Analysis is based on MAESTROv3 latents due to rebuttal time/resource constraints.
>
> **Gaussianity (kurtosis):** We applied z→sign(z)|z|^p to vary kurtosis (up to 20× initial value) while keeping correlations stable. The peak shifts by at most −0.15 proportionally to the kurtosis factor, while the bell shape is fully preserved. Theoretically, non-Gaussianity introduces a residual r(x,λ) reducing effective irreducible variance non-uniformly in λ, shifting the peak.
>
> **Correlations:** We injected inter-channel correlation via z→(1−α)z+α·mean_channels(z). The closed-form $λ^*_F$ breaks down above |ρ|>0.6, with stronger impact than kurtosis. Theoretically, anisotropy causes tr(Σ₁²) in the denominator to be dominated by off-diagonal entries, shifting the prediction toward 0.
>
> **Correlation relaxation:** On pratical scenario, we can relax the isotropy assumption by computing Φ⁻¹ numerically (Proposition 4.1) instead of the closed form. We tested this on our correlated perturbation experiments: the numerically predicted $λ_{irr}$ recovers the correct peak at each correlation level, and the analytical $λ^*_F$ matches it when correlation is low.
>
> **OLS validation:** We confirmed memorization peaks where linear information is minimal. We trained OLS model and compared to Transformer. Transformer/OLS test loss ratio equals 1 at $λ∈{0,1}$ and peaks at $λ^*_F$, consistently across MAESTROv3, MTG-Jamendo, and scaled $Σ_0∈{N(0,I), N(0,0.25I), N(0,4I)}$.
>
> These analyses will be generalized and integrated into an extended Section 4.4.
>
> **Sigmoidal scheduler:** Switching to a sigmoidal scheduler yields a peak increases of 40% on MAESTRO, suggesting scheduler design significantly impacts the memorization signal. This will be added in an appendix.
>
> **W4.** _Gap to large-scale system_
>
> We agree our claims were overstated. As shown in Q3, the peak is observed across various architectures and scales, suggesting the phenomenon extends to larger models. Rebuttal time constraints prevented testing on FLUX/SD3, but it is planned as a next step. Following reviewer's advice, we will reframe the contribution as foundational: characterizing where and why membership signals arise in RF, rather than claiming direct applicability to deployed systems.

---

### Decision · Program_Chairs · 2026-04-30

**Decision:**

Accept (regular)

**Comment:**

The reviewers agreed that the contribution of the work, namely, a theoretical framework for analyzing membership signals in “Rectified Flows” is significant, and their demonstration of Membership Inference Attack on an audio dataset is convincing and useful. There were some concerns about where their assumptions fail for large modern models and datasets. We encourage the authors to elaborate further on this aspect in their final version to ensure the completeness of this work. Overall, this paper seems to make a good contribution to the ICML community, and therefore, I suggest acceptance.